

# Modal sensitivity of rock glaciers to elastic changes from spectral seismic noise monitoring and modeling

Antoine Guillemot[1], Laurent Baillet[1], Stéphane Garambois[1], Xavier Bodin[2], Agnès Helmstetter[1], Raphaël Mayoraz[3], Eric Larose[1]

[1] Univ. Grenoble Alpes, CNRS, Univ. Savoie Mont-Blanc, IRD, IFSTTAR, ISTerre, 38000 Grenoble, France
[2] Univ. Grenoble Alpes, CNRS, Univ. Savoie Mont-Blanc, Laboratoire Environnements, Dynamiques et Territoire de Montagne (EDYTEM, UMR 5204), 73000 Chambéry, France
[3] Canton of Wallis, 1951 Sion, Switzerland

*Correspondence to*: A. Guillemot (antoine.guillemot@univ-grenoble-alpes.fr)

**Abstract.** Among mountainous permafrost landforms, rock glaciers are mostly abundant in periglacial areas, as tongue-shaped heterogeneous bodies. Passive seismic monitoring systems have the potential to provide continuous recordings sensitive to hydro-mechanical parameters of the subsurface. Two active rock glaciers located in the Alps (Gugla, Switzerland and Laurichard, France) have then been instrumented with seismic networks. Here, we analyse the spectral content of ambient noise, in order to study the modal sensitivity of rock glaciers, which is directly linked to elastic properties of the system. For both sites, we succeed in tracking and monitoring resonance frequencies of specific vibrating modes of the rock glaciers during several years. These frequencies show a seasonal pattern characterized by higher frequencies at the end of winters, and lower frequencies in hot periods. We interpret these variations as the effect of the seasonal freeze-thawing cycle on elastic properties of the medium. To assess this assumption, we model both rock glaciers in summer, using seismic velocities constrained by active seismic acquisitions, while bedrock depth is constrained by Ground Penetrating Radar surveys. The variations of elastic properties occurring in winter due to freezing were taken into account thanks to a three-phases Biot-Gassmann poroelastic model, where the rock glacier is considered as a mixture of a solid porous matrix and pores filled by water or ice. Assuming rock glaciers as vibrating structures, we numerically compute the modal response of such mechanical models by a finite-element method. The resulting modeled resonance frequencies fit well the measured ones along seasons, reinforcing the validity of our poroelastic approach. This seismic monitoring allows then a better understanding of location, intensity and timing of freeze-thawing cycles affecting rock glaciers.

## 1 Introduction

Among mountainous landforms, rock glaciers are mostly abundant in periglacial areas, as tongue-shaped heterogeneous bodies. They are composed of a mixture of boulders, rocks, ice lenses, fine frozen materials and liquid water, in various proportions (Barsch, 1996; Haeberli et al., 2006). Gravitational and climatic processes combined with creeping mechanisms lead these glaciers to become active, exhibiting surface displacements ranging from cm/yr to several m/yr (Haeberli et al., 2010). In the context of permafrost degradation associated to climate warming, destabilization processes coupled to an



increase of available materials are increasingly observed in a large range of alpine regions (Bodin et al., 2016; Delaloye et al., 2012; Marcer et al., 2019b; Scotti et al., 2017), thus increasing the risk of torrential flows (Kummert et al., 2018; Marcer et al., 2019a). Therefore, monitoring of active rock glaciers has become a crucial issue to understand physical processes that determine rock glacier dynamics, through thermal, mechanical and hydrological forcings (Kenner et al., 2019; Kenner and Magnusson, 2017; Wirz et al., 2016) and consequently to better predict extreme events threatening human activities. Indeed,

linking internal mechanisms at work to environmental factors remains poorly constrained (Buchli et al., 2018), and lacks quantitative models constructed from high resolution observations.

In this view, rock glacier monitoring is a high-challenging field that has developed over the last decades through several methods (Haeberli et al., 2010). Investigating the internal deformation remains very costly and limited in temporal and spatial scales with geophysical methods and/or borehole investigations (Arenson et al., 2016). Kinematics of the

topographical surface is more accessible by remote sensing methods (with terrestrial photogrammetry or laser scanning, aerial or spatial imagery), together with *in situ* measurements (differential GPS, total station) (Bodin et al., 2018; Haeberli et al., 2006; Kaufmann et al., 2019; Strozzi et al., 2020). However, the knowledge of the medium geometry and composition along with its internal processes that drive rock glacier dynamics require more investigations in depth (Springman et al., 2013). Boreholes provide useful data (temperature, composition, deformation along depth) but remain cost-effective and

limited to one single point of observation. By measuring physical properties sensitive to hydro-mechanical parameters of the medium, a wide range of geophysical methods provides interesting tools to characterize and monitor rock glaciers at a larger scale (e. g. Duvillard et al., 2018; Kneisel et al., 2008; Maurer and Hauck, 2007). However, the need of high resolution temporal monitoring reduces the choice of geophysical methods.

Passive seismic monitoring systems have the potential to overcome these difficulties on debris slope (Samuel Weber et al.,

2018; S. Weber et al., 2018) and permafrost environments (James et al., 2019), as recently illustrated on the Gugla rock glacier (Guillemot et al., 2020). Indeed, seismological networks provide continuous recordings of both seismic ambient noise and microseismicity. The former allows us to estimate tiny seismic wave velocity changes associated to hydro-mechanical variations through ambient noise correlation method, while the latter monitors and locates in time and space the seismic signals generated by rockfalls or by internal cracking and deformation. With these techniques, the seasonal freeze/thawing

cycles have been monitored on the Gugla rock glacier during four years (Guillemot et al., 2020), by quantitatively measuring the increase of rigidity within the surface layers (active and permafrost layers) during wintertime. Seismic velocity drops have also been observed during melting periods, indicating thawing and water infiltration processes occurring within the rock glacier.

The goal of this study is to extend the freeze-thawing cycle observations previously obtained on a single site from seismic

noise correlation (Guillemot et al., 2020) to modal monitoring of two rock glaciers, and evaluate similarities and differences between these two methods. Assuming a rock glacier as a vibrating system, the resonance frequencies that naturally dominate and their corresponding modal shapes should provide information about mechanical parameters of this system. Hence these frequencies and modal parameters are directly linked to elastic properties of the system, which evolve according





to its rigidity and its density (Roux et al., 2014). Since decades, such modal analysis and monitoring of structures have been
performed using seismic ambient noise, especially for existing buildings (Guéguen et al., 2017; Michel et al., 2010) and rock
slope instabilities (Burjánek et al., 2010; Lévy et al., 2010). Both numerical simulations and laboratory experiments have
been already performed with ambient seismic noise sources to confirm the potential of such non-invasive monitoring of
modal parameters changes of a structure, as a bending beam (Roux et al., 2014). Here, we propose to evaluate the potential
of this methodology on two rock glaciers located in the Alps (Laurichard in France and Gugla in Switzerland), at elevations
where climatic forcing dominate the variations of their internal structures and consequently their dynamics. We focus on the
spectral content of continuous seismic data (noise and earthquakes) to track and monitor resonance frequencies. Our goal is
to detect vibrating modes of the rock glacier, and the time variability of their resonance frequencies gives hints to better
quantify and locate the changes of rigidity resulting from freeze-thawing effects on surface layers. These observations have
been numerically modeled using a finite-element method, towards a mechanical modeling of such rock glaciers.

After presenting the two studied rock glaciers and their instrumentation, we present the methodology to perform a spectral
analysis from seismic data, and the resulting resonance frequencies variations observed on both sites. In a second part, we
detail the mechanical modeling of those rock glaciers, based on finite-element method and constrained by several
geophysical investigations, which allows to compute synthetic resonance frequencies and to understand their sensitivity.
Finally, we compare observed and modeled modal studies, in order to converge to a consistent view of those rock glaciers
and their freeze-thawing cycles.

## 2 Presentation of the sites

### 2.1 The Laurichard rock glacier

#### 2.1.1 Context

As presented in previous studies (Bodin et al., 2018, 2009; Francou and Reynaud, 1992), the Laurichard catchment was
chosen as a test site for different geomorphological studies conducted since several decades. This large thalweg is part of the
Combeynot massif, which is a crystalline subsection of the Ecrins massif located in the South of Lautaret pass. This area
constitutes a climatic transition between northern and southern French Alps. Several rock glaciers are observed in this area in
different states of deformation, from relicted to active ones. Among them, the Laurichard rock glacier is the most studied
active rock glacier in the French Alps ((Bodin et al., 2018; Francou and Reynaud, 1992), Fig. 1a). It appears as a 800 m long
and 100-200 m wide tongue-shaped landform of large boulders, flowing downstream between the rooting zone (2650 m a.s.l)
and the front zone (2450 m a.s.l). It is fed by the gravitational rock activity that originates on the surrounding slopes,
composed of highly fractured granitic rockwalls providing large boulders ($10^{-2}$ – $10^{1}$ m diameter). It shows rather simple and
evident features of active rock glacier morphology (Bodin et al., 2018): transversal ridges and furrows, steeper lateral talus





and rock activity at the front, unstable rock mass on the surface. These geomorphological hints typically reveal creeping
movement of the whole debris mass, with the presence of ice mainly responsible for this rock glacier dynamics.

The kinematic behavior of the Laurichard rock glacier has been studied since several decades: large blocks have regularly
been marked since 1984 (Francou and Reynaud, 1992) together with other remote sensing techniques or geodetic
measurements (total station, Lidar and GPS). The long term survey permits to measure surface velocity with different
temporal and spatial scales, reaching very high resolution (below one day and one meter, (Marsy et al., 2018)). The general
spatial velocity pattern shows a main central flow line along the maximal slope. At the frontal zone, it appears that the right
orographic side is the most active part of the rock glacier. The mean annual surface velocity of the site is measured at around
1m/yr, with a progressive increase between 2005 and 2015, probably in reaction to the observed increase in mean annual air
temperature during this period (Bodin et al., 2018). This latter acceleration has been observed synchronously on other
monitored rock glaciers in the Alps (Delaloye et al., 2008; Kellerer-Pirklbauer et al., 2018) and is most probably resulting
from the warming of the permafrost (Kääb et al., 2007).

### 2.1.2 Available data and knowledge

The thermal regime of the rock glacier is monitored since 2003 thanks to Miniature Temperature Dataloggers (MTD) that
record every hour the temperature of the sub-surface (below 2-10 cm of debris) at five different locations (one being located
outside the rock glacier, see Fig. 1a). Geophysical investigations have been performed several times since 2004, especially
with ERT surveys providing a first estimation of the internal structure (ice content and thickness) subject to permafrost
degradation (Bodin et al., 2009).

The topography of the rock glacier has also been regularly surveyed since two decades using high-resolution Digital
Elevation Models (DEM) computed from terrestrial and remote-sensing methods (Bodin et al., 2018). In this study, a DEM
at a 10 m resolution derived from the IGN (French national institute for geographic information) BD Alti product was used.
Additionally, a DEM of the bed over which the rock glacier is flowing was interpolated from manually drawn contour lines
based on surface DEM. These contour lines of the bed extend the contour lines of the terrain surrounding the rock glacier
below its surface, using local constraints from existing geophysical data ((Bodin et al., 2009). For this operation, we assume
that the rather simple overall morphology of the Laurichard rock glacier (a single relatively narrow tongue) and its
overimposed position above surrounding terrain (bedrock and other debris slopes, called "bedrock" below) allow to estimate
the lateral thickness variability of the rock glacier. This DEM of the bedrock is coherent with bedrock depth derived from
GPR acquisitions conducted in 2019, and thus will be used to constrain Laurichard rock glacier geometry (see section 4.2.1).



### 2.1.3 Passive seismic monitoring

Since December 2017, an array of six seismometers (named C00 to C05) has been set up on the lower part of the Laurichard rock glacier (Figure 1e). They are located around 50 m apart, covering the whole area of the rock glacier front. The seismometers (Mark Products L4C) are coupled with the top of relatively large, stable and flat boulders, and sheltered by a plastic tube to shield off any influence of rain, wind and snow. They are connected together to one digitizer (Nanometrics Centaur, sampling rate 200 Hz) with wires insulated by sheath to protect from weather and rockfalls. This passive

seismological network records continuously ambient noise together with microseismicity.

Because of a rough field context (climatic conditions and surface instability, subjecting sensors to tilting), and although frequently required site visits, the long periods of usable data have been recorded from only two seismometers : C00 (located around 100 m upstream the front, on the right side of the rock glacier) and C05 (located near the front, on the left side). Therefore we decided here to present results from only these two locations. In the following part, we used these passive

seismic recordings to model the dynamic response of the site, through their spectral analysis. Other sensors, though discontinuously active, and not presented here, yield the same observations and conclusions when in operation.

### 2.2 The Gugla rock glacier

Located in Wallis Alps, the Gugla rock glacier (also called Gugla-Breithorn or Gugla-Bielzug) is part of a large number of

active rock glaciers that have been regularly investigated over the last decade in this geographic area (Delaloye *et al.* 2012, Merz *et al.* 2016, Wirz *et al.* 2016, Buchli *et al.* 2018). Ranging from 2550 to 2800 m *asl*, its tongue-shaped morphology covers about 130 m in width, 600 m in length, and is up to 40 m thick in its downstream part. Since 2010, surface velocities have been measured about 5 m/yr at the front, with a peak in the southern part culminating, in 2013, at a velocity of more than 15 m/yr. This increase in velocity has also propagated to the rooting zone (from 0.6 m/yr in 2008 to 2 m/yr in 2018, as

evidenced by geodetic measurements). Debris detachments from the rock glacier front supply yearly one or more torrential flows triggered from an area located immediately downslope of the rock glacier front, regularly reaching the main valley downstream with dense human facilities. Hence, the risk of runout onto the village of Herbriggen and railways and roads nearby remains high after intense snowmelt or following long-lasting or repeated rainfall, involving volumes from 500 to more than 5000 $m_3$ per event (Kummert and Delaloye, 2018).

In addition to meteorological stations and GPS monitoring systems, a seismological network has been set up since October 2015, covering the lower part of the rock glacier (Guillemot et al., 2020). It is composed of five seismic sensors (labelled C1 to C5, Sercel L22 geophones with a resonance frequency of 2 Hz), including two of them (C2 and C4) located on the glacier's longitudinal axis, whereas the others are placed on the two stable sides (Figure 1e).

In addition, eight boreholes and one geophysical campaign from seismic refraction were performed on the rock glacier in

2014 ( Geo2X, 2014, CREALP, 2016, 2015) in order to better constrain the internal structure of the subsurface (thickness





and composition of the layers, seismic velocity). Through two thermistor chains that have continuously recorded temperature at depth (until 19.5 m) between 2014 and 2017, the active layer thickness has been located at around 4.5 m (+/-20%) (CREALP, 2016). Finally, three webcams provide hourly images showing different viewpoints of the rock glacier front (Kummert et al., 2018).

**3 Spectral analysis of seismic data**

**3.1 Methods**

Continuous seismic monitoring are autonomous operating systems composed of an array of seismometers that permanently record particle vibrations on the ground related to microseismicity and noise. Microseismicity is increasingly used for precisely locating the seismic signals induced by mass movements, avalanches and rockfalls ( Spillmann et al., 2007, Amitrano et al., 2010; Helmstetter and Garambois, 2010; Lacroix and Helmstetter, 2011). Ambient seismic noise is used to investigate the medium between several sensors, and monitor subsurface properties variations (Snieder and Larose, 2013, Larose et al, 2015, for a review).

Experimental results combined with numerical modeling showed that resonance frequencies of a structure can be derived from the spectral analysis of ambient seismic noise recorded on it (Lévy et al., 2010; Michel et al., 2010). Different applications were successfully proposed: as a monitoring method for a prone-to-fall rock column (Lévy et al., 2010) or as a way of tracking dynamic parameters of existing structures. Indeed, ambient vibrations provide information about the modal parameters of a structure, defined as resonance frequencies, modal shapes and damping ratios (Michel et al., 2008). These features can be deduced from the frequency content of seismic recordings, which depends on source and propagation properties, but also on structural, geometrical and elastic properties of the structure (Roux et al., 2014). The seismic noise averaging property allows considering the illuminated frequency spectrum as large and stable enough to overcome source and trajectories effects, particularly when monitoring is considered. The power spectrum density (PSD) is simply defined by averaging the intensity of the fast Fourier transform (FFT) of the seismic record $\varphi_a(t)$ : $PSD(\omega) = |FFT(\varphi_a(t))|^2$, where $\omega$ is the angular frequency. In resonant structures like sedimentary basins, rock columns, mountain slopes and buildings, high peaks in the PSD could correspond to specific vibration modal shapes of the structure (Larose et al., 2015). The corresponding frequencies, identified as resonance ones, mainly depend on geometrical (characteristic length, cross-section, shape), structural (boundary conditions) and mechanical (density, Young's modulus) features defining the structure.

As an example, one can approximate a (soft) sedimentary cover overlying a (hard) bedrock using a 2-D semi-infinite half-space covered by a soft layer of density $\rho$, thickness $h$, average shear-wave velocity $V_s$ and shear modulus $\mu$. Such simple mechanical modeling leads to the well-known analytic solution of the first resonance frequency $f_0$ corresponding to the fundamental mode (Parolai, 2002): $f_0 = \frac{V_s}{4h} = \sqrt{\mu/\rho}/4h$ . Extending this model to a rock glacier shows that in absence of geometrical changes, resonance frequency variations can be related to evolution of its rigidity, through Young's modulus and



of its density. In this study, our goal is to track the temporal evolution of resonance frequencies of a rock glacier, considering it as a vibrating structure, in order to understand their physical causes and then to monitor any variation of mechanical properties.

To compute the PSD, we pre-processed hourly raw seismic traces with: i) instrumental response deconvolution, demeaning and detrending; ii) clipping (high-amplitude removal by setting a maximum threshold equal to four-times the standard deviation). Then we computed PSD using Welch's estimate (Solomon, 1991) between 1 and 50 Hz (with Tukey windowing, 10% overlapping and 4096 points for discrete Fourier transform and then hourly-averaging and normalization by the hourly maximum). We then obtained hourly-normalized spectrograms, containing the relative weight of each frequency. We pick

local maxima automatically, by using different threshold values (minimum of frequency at 10Hz, minimum of peak height, minimum of peaks distance, minimum of prominence and maximum of width), in order to automatically track and monitor sharp peaks. We selected only frequency peaks above 10 Hz in order to prevent any source effects, since specific peaks of the rock glacier are assumed to be above this limit (see Figure 2).

For the Laurichard site, we used seismic traces of another station located in a stable area at Lautaret pass (see Figure 1b),

named OGSA (RESIF, 1995). Since OGSA is considered as a reference station, we could compare spectral contents with the reference one, to evaluate the specific frequency peaks of the Laurichard rock glacier (see Figure 2). In this way, we ensured that those picked frequencies are related to the modal signature of the rock glacier. Since no sensor was settled out of the rock glacier in Gugla site, we have not applied this method for this site.

We also applied the same method for earthquakes signals, but the results appear less clear than those from ambient noise (but

shown and discussed in Appendix A).

## 3.2 Resonance frequency monitoring of Laurichard rock glacier

The results of Power Spectral Density (PSD), normalized every hour between 1 and 50 Hz, are shown for the two seismometers C00 and C05 in Figure 3. Several peaks of PSD appear and vary along time.

Among potential sources affecting the spectral content of seismic records, we aimed at selecting only natural resonance

modes of the rock glacier structure. For example, we observed a very stable narrow peak of PSD at 23 Hz for both seismometers. This mode lights up mainly during summertime and in the daytime, although it remains visible during winter and in the nighttime, but with significantly lower amplitude. Since this frequency peak is strictly stable in time and no subglacial resonating water-filled cavities was known in this site (Roeoesli et al., 2016), we interpret its origin as anthropogenic, possibly generated by a pressure pipe located 400 m downstream or from road traffic coupled with a tunnel

near the Lautaret pass (see Figure 1). This frequency peak is also visible from spectrograms of station OGSA (see black arrow in Figure 2d) located at Col du Lautaret on a stable site, reinforcing the assumption of its anthropogenic source. The spectral content of these recordings exhibits the same peak at 23-24 Hz (see red curve in Figure 2d), suggesting it is not related to the rock glacier. This frequency peak is hereafter excluded.





Other spurious effects of artificial or non-specific sources affecting PSD are known: atmospheric effects (local structure or vegetation coupled with wind (Johnson et al., 2019)), loss of sensor coupling or water filling of the resonating structure of the sensor during melt out (Carmichael, 2019). However, these sources are not present at Laurichard rock glacier: for all sites the seismometers are well-coupled on flat and stable boulders, ensuring a good rock-to-sensor coupling. Each of them is sheltered by a plastic tube covered by a waterproof tarp, in order to prevent any influence of rain, wind and snow. During site visits, no water in the settlement was observed.

For the C00 seismometer, we observe a main peak of PSD between 15 and 20 Hz interpreted as the fundamental mode of the nearby area of the rock glacier (Figure 3a). The temporal evolution of this mode shows a seasonal cycle, characterized by higher frequencies during winters and lower frequencies during summers. A sudden drop of frequency occurs at the time when melting processes occur (blue boxes on Figure 3). Comparing the two winter periods, the maximum frequency is lower in 2019 (around 17 Hz) than in 2018 (around 19 Hz), while it remains constant around 15 Hz for the two recorded summers.

For sensor C05, we can follow the same peak considered as the fundamental mode of the corresponding area, with a similar seasonal cycle (Figure 3b). Again, the fundamental frequency increases during winter, and drops during melting periods and summer time. Compared to the C00 case, the amplitude of this seasonal variation is much higher: even if the frequency value in summer is similar (around 15 Hz), the winter one reaches a higher value (around 30 Hz). The maximum value is also higher in 2018 (35 Hz) than in 2019 (30 Hz).

### 3.3 Resonance frequency monitoring of Gugla rock glacier

We applied the same spectral analysis for the Gugla site. From the hourly normalized PSD of seismic noise recorded on the rock glacier (sensor C2), we observed two resonance frequencies evolving with time (Figure 4). At relatively high frequencies, a second mode is well measured, because the mean noise level is higher in Gugla than in Laurichard, where only the fundamental mode is observed.

As for Laurichard site, these frequencies present seasonal oscillations: they increase progressively to peak at cold winter periods, whereas they drop when melting processes occur at summer times (blue boxes in Figure 4). The fundamental mode varies from 15 Hz in summertime to around 21 Hz in wintertime, whereas the second mode oscillates from 27 Hz to 40 Hz.

Again, the resonance frequency of the fundamental mode shows an inter-annual variability: in winter 2017 the maximum value is lower (around 20 Hz) than the peaks of the two other winters (around 24 Hz).

## 4 Mechanical modelling

### 4.1 Methodology

Using a finite-element method, we model rock glaciers as vibrating structures embedded in the bedrock. We then study the sensitivity of the modal response of this model to ambient seismic noise as a function of its elastic properties.

Elastic features can be determined as a function of compressional $V_p$ and shear $V_s$ seismic wave velocities, together with the density $\rho$. Therefore we evaluate seismic velocities along depth thanks to active seismic investigations complemented by Ground Penetrating Radar (GPR) surveys in order to obtain a 1D $\left[V_p(z), V_s(z), \rho(z)\right]$ profile describing the medium near the seismometer of interest. This first model is considered as a reference model since it has been built during unfrozen summer periods. In addition, we consider the effect of freezing-thawing processes on the elastic model using a poroelastic approach

that enables to quantitatively evaluate elastic parameter changes due to the freezing. Modal analysis is then performed with Comsol software1, in order to compute synthetic resonance frequencies that can be compared with the observed ones.

## 4.2 Reference model from geophysical investigations

Since one decade, numerous experiments have been devoted to geophysical characterization of rock glaciers (Maurer and Hauck, 2007; Kneisel et al., 2008; Haeberli et al., 2010) in order to constrain site modeling and better understand subsurface

physical processes involved in their deformation. Among available geophysical methods, seismic refraction tomography (SRT), Ground Penetrating Radar (GPR) and Electrical Resistivity Tomography (ERT) have provided promising results. In alpine permafrost regions, the high heterogeneity of the subsurface together with cost-effective and risky field conditions make geophysics challenging. However, combining the geophysical methods listed above give useful information in a view of imaging and modeling the subsurface.

### 4.2.1 Laurichard rock glacier model

#### 4.2.1.1 Ground Penetrating Radar survey

We performed a Ground Penetrating Radar (GPR) campaign at the end of June 2019 to better assess the geometry and the internal structure of the Laurichard rock glacier. It is composed of (i) a common-offset longitudinal profile starting in the

middle of the rock glacier and stopping near the front, following the main flow line and (ii) a common-offset transversal profile crossing over the rock glacier width, approximately following the C01-C04 seismometers line discussed afterwards (Figure 1a).

Preliminary tests have demonstrated the ability of the 25 MHz Rough Terrain Antennas (RTA) to follow the continuity of the reflectors throughout the glacier, despite lower resolution (wavelength about 4.8 m). The 100 MHz antennas actually

experienced penetration problems, presumably related to the presence of heterogeneities equivalent in size to the wavelength (about 1.2 m). In addition, a Common Middle Point (CMP) survey was performed along the western part of the transverse profile using unshielded bi-static 100 MHz antennas, in order to assess locally the electromagnetic wave velocity distribution within the glacier. Fig. 5(1a) shows the CMP data after trace by trace amplitude normalization and gain amplification using a dynamic automatic gain control computed on a 100 ns time window. After the direct air and

1 https://www.comsol.fr





ground waves, numerous events exhibiting a hyperbola shape can be recognized from 40 ns to 225 ns in the CMP data. These hyperbolas have been analyzed considering a semblance analysis (Fig. 5(1b)), which yields the stacking velocity versus propagation time where a semblance is maximum. The picked maximum of the velocity distribution shows variations ranging from 14 cm/ns to 11 cm/ns with a mean velocity of 12 cm/ns. As these variations are measured on apparent velocities, the real variations are larger when layers are considered. They can be qualitatively interpreted in terms

of an increase of air (velocity of 30 cm/ns) and ice (velocity of 17 cm/ns) content when velocity is large, and an increase of water content (velocity of 3.33 cm/ns) when velocity drops. Considering a mean velocity of 12 cm/ns, the 100 MHz CMP analysis penetrates to a depth of 13.5 m and the increase of velocity arriving nearby 110 ns corresponds to a depth around 6 m.

        Figure 5(2) shows both common-offset profiles acquired using the 25 MHz antennas after they were processed using: i)

time-zero source correction, ii) normal-moveout correction as source and receivers are separated by an offset of 6.2 m for these antennas, iii) static corrections for topography iv) migration and v) time to depth conversion. The later processing steps have been performed considering a mean velocity of 12 cm/ns, a value deduced from the CMP analysis (Fig. 5(1b)).

        Both GPR images show relatively continuous reflectivity within the rock glacier, particularly along the longitudinal direction, indicating a stratification of the deposits. The use of low frequency antenna certainly naturally homogenized the

heterogeneity of rock glaciers, as witnessed by the quasi-absence of diffraction. The thickness of the glacier varies weakly along the longitudinal direction, ranging from 28 m upstream to 10 m downstream. More abrupt variations are detected in the transverse direction (Figure 5(2b)), from a few meters to 20 m at the center and the eastern part. It must be noted that the first few meters of the rock glacier cannot be resolved, due to the RTA antennas configuration with a large source-receiver offset and the large wavelength (about 4.8 m).

As a conclusion, the bedrock interface depth is well constrained by GPR results, combining longitudinal and transversal profiles. In the lower part of the rock glacier near the front, the bedrock is estimated at around 10 m depth. But the transversal profile also reveals heterogeneities over the seismic array. In the western part (C05) the rock glacier seems thinner than in the eastern part (C00), according to the bedrock depth estimation based on contour line interpolation on both sides of the rock glacier. By Digital Elevation Model (DEM) difference between surface and bedrock (see 0), we then more

precisely estimated the interface depth (14 m for C00 and 8 m for C05, see Figure 8(b)).

### 4.2.1.2 Seismic tomography

        A seismic refraction/tomography survey has been performed in July 2019. This experiment consists of active seismic

recordings with controlled sources, in order to determine the P-wave velocity distribution along a 2D line. The profile composed of 24 geophones (4.5 Hz) deployed every 3 m is roughly located along the C1-C4 line, near the center of the seismic array (Figure 1a). The first arrival time picking of the 8 shots have been inverted using a Simultaneous Iterative Reconstruction Technique (SIRT, (Demanet., 2000)) in order to obtain the P-wave velocity distribution along the profile





(Figure 6). From an initial model with a uniform velocity of 3000 m/s (340 m/s in the air), 25 iterations were performed to

reconstruct observations ($RMS\ misfit = 8\ ms$). The result shows 2D variations with some degree of layering in the velocity distribution. The interface between the rock glacier and the bedrock might be marked by the large interface separating a material with velocities lower than 2000 m/s with a layer showing a large velocity about 3000 m/s. Its thickness varies from 10 m to 20 m, which is consistent with GPR results (Figure 5(2)). To overcome the smoothing effect of seismic tomography, data have also been processed using seismic refraction with two opposite large-offset shots. This approach highlights a

layered structure of the medium, with different slopes and particularly an interface located around a depth of 4 m, which probably separates the active layer from the permafrost one. Therefore, we can assume an active layer from the surface to 4 m depth, corresponding to the maximal depth where the medium is totally thawed in summertime.

### 4.2.1.3 Multichannel Analysis of Surface Waves (MASW)


In order to better understand the seismic wavefield and constrain the S-wave velocity distribution at the site, we analyzed the surface Rayleigh waves, which dominate the vertical seismic records used in the tomography. For this, we used a far offset shot and computed the semblance map of the velocity and frequency of the waves dominating the seismic record (Figure 7a), obtained using the Geopsy package (Wathelet et al., 2004).

The semblance map shows several continuous modes, while the fundamental dispersion curve was picked from 14 Hz to 30 Hz, as indicated by the black line. The presence of several other modes is due to the presence of strong contrasts within the rock glacier and at the interface between the rock glacier and the bedrock. The dispersion curve was inverted using the Geopsy/dinver package (Wathelet et al., 2004), where a global neighborhood algorithm optimization method is implemented. The model was parametrized using four layers, the top three searching for linear velocity gradients in each layer. With the

available frequency range and the velocity distribution, the resolution at large depths (> 15 m) is rather poor. The Vs profiles displayed in Figure 7b shows a large variability but the best fitting models all converge towards an interface located at 5 m depth with a superficial velocity of 155 m/s followed by a linear increase of velocity reaching 750 m/s at a depth of 7 m. The best model also shows another deeper interface, at 15 m depth, which could be the bedrock interface, despite the low resolution at this depth.

From all these geophysical surveys, a tentative 1D seismic velocity model was built for each seismometer (C00 and C05), as the reference unfrozen model. Its values have been well constrained by seismic refraction, whereas bedrock interface depth has been constrained by GPR results, together with interpolated DEM differences (Figure 8).

### 4.2.2 Gugla rock glacier model

To establish a reference model of the Gugla rock glacier, we use seismic velocities that have already been constrained by a

seismic refraction survey (Fig. 1b) performed in July 2017 during a summer and dry period (Figure 9). All values for Vp and



Vs profiles have been already presented in a previous study (Guillemot et al., 2020). We also assume a density profile that progressively increases, from $\rho = 2000 \, kg/m^3$ at the surface to $\rho = 2800 \, kg/m^3$ at the bedrock.

Moreover, we estimate the bedrock at 23 m depth, in accordance with observations provided by boreholes located near the seismometer of interest (borehole F2, (CREALP, 2015)).

### 4.3 Freezing modeling from poroelastic approach

#### 4.3.1 Methods and results

In order to mimic the freeze-thawing effect on resonance frequencies, the associated variations of elastic properties of the material have to be constrained by seismic velocity changes. A winter model is required to be compared to the summer one. For the transition from summer to winter, an increase of P- and S-wave velocities during winter is expected, in accordance with laboratory and numerical experiments (Timur, 1968; Carcione and Seriani, 1998; Carcione et al., 2010). Indeed, both bulk and shear moduli of the effective medium increase during freezing, generating a global stiffening of the upper part of the rock glacier subject to the seasonal thermal forcing.

In order to quantify the evolution of these elastic parameters with freezing, we use a poroelastic approach assuming a three-phase model: a rock glacier is considered as a porous material composed of pores embedded into a granular rocky matrix. We then address the sensitivity of elastic parameters to the proportion of liquid water and ice filling the pores, for several porosity values. Since the wavelength of seismic waves is much greater than the size of the pores, this homogenization approach holds. As did Carcione and Seriani (1998), we use a Biot-Gassmann type three-phases model that considers two solid matrices (rock and ice) and a fluid one (liquid water). Since the contribution of air proportion within the pore is negligible on the shear modulus, which mostly determines the fundamental vibrating mode, we omit the air phase for the sake of simplicity.

We apply the following methodology for the three cases (Gugla C2, Laurichard C00 and C05). Several parameters are required to completely describe the poroelastic state of a rock glacier: bulk and shear moduli of the respective pure phases, the averaged density, the porosity and the water saturation.

For the summer state of the rock glacier, we evaluate these parameters indirectly. The density $\rho$ is fixed at realistic values ($\rho = 1800 \, kg/m^3$ for the first two meters, $\rho = 2000 \, kg/m^3$ for the deeper part of the rock glacier, and $\rho = 2650 \, kg/m^3$ for the bedrock, (Hausmann et al., 2012)), as well as the porosity profile (see references in paragraph 0). The water saturation $s$ is assumed at $s = 0$ for the first two meters, and $s = 0.2$ deeper, in consistency with visual observations and qualitative features performed from boreholes in summer (CREALP, 2016). Respective bulk and shear moduli of the pure phases (ice and water) are fixed from the example of Berea sandstone ((Carcione and Seriani, 1998), Table 2). The bulk and shear moduli of the dry solid matrix have been obtained from the inversion of velocities using a Biot-Gassmann poroelastic model with two phases (solid matrix and water). For this inversion step, we use water saturation, porosity and seismic velocity profiles ($V_p$, $V_s$) deduced from seismic refraction geophysics performed in summer (see 0 for Laurichard and 0 for Gugla).



The outputs of this inversion step are the elastic moduli of the solid matrix, assumed constant along seasons, and describing the elastic behaviour of the rock glacier without neither water saturation nor seasonal freezing. The profile of all the

parameters of this summer model are shown in Figure 10(b,c,d,e) in red curves.

For the winter state of the rock glacier, we keep unchanged the porosity and elastic parameters of the three phases (water, ice, and solid matrix), but we assume the pores are fully filled by ice (water saturation equals to zero) from the surface to the maximum depth where seasonal freezing acts, also called Zero Annual Amplitude (ZAA) depth. ZAA is estimated to approximately 8 m depth from thermal investigations in Gugla ((CREALP, 2016)), and extrapolated as well to Laurichard.

The averaged density is computed by averaging the density of each phase, weighted by their respective volumetric ratio.

The seismic velocity profiles ($V_p$, $V_s$) for a totally frozen state are then computed by appling the 3-phases poroelastic model. For this step, the input parameters are the porosity and the density, together with bulk and shear moduli of each phase (water, ice and solid matrix). These elastic parameters are homogeneized according to Carcione and Seriani, 1998, and then equations of wave propagation are solved in order to obtain fast P-wave and S-wave velocities as modeled by Leclaire et al.,

1994. Results of the evolution of these velocities with respect to ice/water ratio filling the pores are shown in Figure 11 for the example of Laurichard (sensor C00). Hence we deduce the values for a frozen state of the rock glacier with pores totally filled by ice between the surface and ZAA. We acknowledge that this is a strong assumption for the winter state, and that other models may also explain our observations. The profiles of all the parameters of this winter model are shown in Figure 10(b,c,d,e) with blue curves.

With these two models in summer (minimum of freezing) and winter (maximum of freezing), we can also model the transition between them. Although the freezing process (from summer to winter) is poorly constrained, due to liquid water infiltration and complex thermal forcing, the thawing process (from winter to summer) appears easier to model, assuming a temporal evolution of thawing mainly controlled by thermal heat wave propagating from the surface to the ZAA depth. Hence, we build an intermediate state of the rock glacier by introducing another parameter, called "maximum depth of

thawing" (see Figure 10a). This parameter establishes an interface between the unfrozen state (as the summer model) above it, and the frozen state (total pore filling by ice) below it. Hence, this maximum depth of thawing evolves from the surface to the ZAA with 1 m increments, reporting as many intermediate models. The profile of the parameters of an example of intermediate model are shown in Figure 10(b,c,d,e) in purple dashed curves.

Finally, we compute the modal response (explained below in section 0) of the corresponding vibrating structure of all these

models (summer, intermediate and winter), modeling a value of resonance frequency depending of the freezing state of the rock glacier.

### 4.3.2 Influence of the porosity

Defined as the ratio between pore volume and total volume, the porosity $\phi$ of the rock glacier is one of the key parameters influencing the mechanical modeling. Our three-phase poroelastic model actually considers the filling of pores by two





phases (ice and water), together with interaction between ice and rocky debris matrices that strongly depends on porosity. In the absence of any *in-situ* information, we assume a model of spherical particles stacking (Rice, 1993), decreasing with depth due to compaction ($\phi = 0.35$ for sublayers from the surface to 2 m depth and $\phi = 0.25$ elsewhere below, for both sites). In order to quantitatively assess the sensitivity of our results to porosity, we also apply the mechanical modeling to other profiles considering extreme values (low limit: $\phi = 0.2$ in the active layer and $\phi = 0.15$ elsewhere, and high limit for

this rock glacier lithology (Arenson and Springman, 2005): $\phi = 0.6$ everywhere). As expected, the higher porosity values, the higher the influence of the ice pore filling on the elastic parameters, and thus the higher the variation of modeled resonance frequencies. Then in the following results presented below, errorbars correspond to the sensitivity on the porosity (low limit for low porosity, high limit for high porosity, see values in transparence in Figure 13).

**4.4 Modal analysis and frequency response of the rock glacier**

We build a mechanical model on Comsol software based on the finite-element method, in order to numerically compute its resonance frequencies and modal response (see Appendix B for details). The rock glacier is modeled as a 2-D rectangular vibrating structure embedded in the bedrock (or a stable bottom layer). The height $H$ of the structure is fixed at the corresponding depth of bedrock (see Figure 8 for Laurichard and Figure 9 for Gugla). The model is vertically sub-sampled into 2 m thick sublayers, with elastic parameters interpolated from averaged values of seismic tomography results, and with a

usual isotropic attenuation factor of 1% (Bonnefoy-Claudet et al., 2006). Depending on the direction of the model (longitudinal or transversal), the width of the structure varies in accordance with the whole rock glacier size (several tens of meters), permitting an infinity of vibration modes. Based on a polarization analysis from ambient noise between 1 and 50 Hz in the Gugla rock glacier in summer 2016, we observed that the wavefield of rock glaciers is mostly polarized in a parallel to the slope direction. Similar to the fundamental mode of an unstable rock mass (Burjánek et al., 2012), the measured

polarization is almost linear (ellipticity lower than 0.15) and thus corresponds best to shear modes. We then computed the Frequency Response Function (FRF) (Fu and He, 2001) on the whole length (several hundreds of meters for both cases) of the rock glacier, in order to obtain its resonance frequencies corresponding to vibration modes of the mechanical structure. For this step, we simulated several seismic sources located at the base of the vibrating structure (see red crosses in Figure 12a), producing harmonic forces from 1 to 50 Hz in all directions. The amplitude of this modeled seismic noise is not

frequency-dependent, while it decreases generally with frequency on the field, showing probably the excitation of other modes than the recorded ones (especially in Laurichard). However, after checking that resonance frequencies obtained from FRF with high amplitude of vertical displacement (Figure 12b) would not be modified, we reduced the width of the model to 5 m by applying symmetrical conditions at the boundaries perpendicular to the slope (Figure 12c), in order to facilitate the following parametric modal analysis.



## 4.5 Comparison between observed and modeled resonance frequencies


We show the results from the modal analysis on Comsol software for only the observed modes with a vertical component (first mode for Laurichard, the two first modes for Gugla) (as shown in Figure 13 for Laurichard, and not presented here but similar for Gugla). Nine mechanical models have been tested, corresponding to different steps of thawing with elastic parameters selected as described in 0. Thus, we present modeled resonance frequencies with respect to the maximal depth of

thawing, from the surface to 8 m depth (Figure 13), and compare them to the maximum values of the observed ones (Figure 3 for Laurichard and Figure 4 for Gugla). The modeled resonance frequencies fit well the observed ones, considering error bars related to porosity uncertainties (see 0). Resonance frequencies of these modes match the frequency band of measurements below 50 Hz, and generally decrease with thawing. Focusing on the fundamental mode (mode 0), the resonance frequency is of the same order of magnitude (between 15 and 20 Hz) for all cases. Focusing on the two sensor

locations in Laurichard, a stronger effect of freezing is observed for C05 than for C00 model. Thus, these numerical results explain well the seasonal variations of observed resonance frequencies, assuming a thawing process from the surface to 8 m depth between winter and summer.

## 5 Discussion

From the results of mechanical simulation on both Laurichard and Gugla rock glaciers, we draw several conclusions:

-   The vibrating modes of rock glaciers can be tracked from spectrograms of seismic ambient noise. The resonance frequencies from the mechanical modeling fit well the measured ones (between 15 Hz and 20 Hz in summer for both sites) within experimental errorbars. This validates our methodology based on rock glacier modeling as a vibrating structure, at least for the first mode;

-   Monitoring these resonance frequencies along time allows to observe seasonal evolution: all the modes show a

progressive increase of the resonance frequencies during winter, followed by a sudden drop in melting periods and lower values during summers.

-   According to the poroelastic approach used to model the effect of freezing on seismic velocities, this variation is qualitatively well explained by freeze-thawing processes. Indeed, the annual heat wave propagates into the surface layers of the rock glacier (Cicoira et al., 2019; CREALP, 2016), causing a change of frozen material content within

the porous medium, and thus a large variation of elastic properties due to this thermo-mechanical forcing. For both sites and sensor locations, this modeled mechanical forcing provides a good estimation of the observed seasonal frequency variations, quantitatively. The modeled changes of elastic parameters (bulk and shear moduli increasing through seismic velocities) involved for Gugla rock glacier (Guillemot et al., 2020) have thus been improved by this complementary method based on a more complete description of poroelasticity, though other models may also

explain our observations.





- By tracking resonance frequencies, we are able to detect inter-annual climate variability. Indeed, the freezing process appears to strongly depend on annual climate variability: as an example, in 2019 in Laurichard, the winter resonance frequency is lower than in 2018, indicating a lower rigidity of the medium due to reduced frozen material content. The winter was actually colder in 2018 than in 2019: from a meteorological station near the col du Lautaret (1 km from Laurichard), the mean air temperature during snow cover $T_{winter}$ was lower in 2018 ($T_{winter}(2018) = -2.07\,°C$) than in 2019 ($T_{winter}(2019) = -0.50\,°C$). The intensity of freezing is generally estimated from Freezing Degrees Day  (FDD), defined as a time cumulative sum of each ground surface temperature below 0°C recorded during one wintertime. In addition to an earlier snow cover period in 2019 than 2018 that insulates the ground from the air forcing, the internal freezing of the rock glacier was less intense in 2019 ($FFD(2019) = -322\,°C.day$) than in 2018 $FFD(2018) = -451\,°C.day$). For the Gugla site, the winter resonance frequency was significantly lower in 2017 than in the others years. Despite a comparable mean air temperature between 2016 and 2017, the earlier and longer snow cover period in 2017 promotes a lower freezing of the internal layers. Similarly, we conclude that resonance frequency in wintertime indicates well the intensity of freeze-thawing effects on the rock glacier.

- Despite a high level of heterogeneities within rock glaciers, low-frequency GPR results allow to better constrain the bedrock interface depth. For Laurichard, the mean value was estimated at 10 m (+/-50% due to the underneath slope). According to field observations and DEM interpolation, we fixed this value at 14 m for C00 model, and 8 m for C05 model. This unique difference between the two locations explains very well the observed gap of seasonal resonance frequency amplitude (Figure 3): the shallower the bedrock interface, the larger this amplitude. In addition to active seismology allowing to perform 2D seismic velocity tomographies, low-frequency GPR results provide valuable information about internal structure of the surveyed rock glaciers, reinforcing the benefits of geophysical investigations in accordance with passive seismology in rock glaciers.

Furthermore, the relation between ground surface temperature and resonance frequencies is plotted in Figure 14 (for Laurichard C00 case). It reveals an annually repeated pattern showing a hysteretic behavior. This non-linear relation suggests several phases over the year (indicated with colors and numbers in Figure 14b), depending on the state of freezing of the rock glacier: 1) dry and unfrozen phase (late summer and autumn), when temperature is varying above 0°C while resonance frequency stays at its lowest level ; 2) shallow freezing phase (late autumn and early winter), when temperature decreases below 0°C (with possible significant drops depending on the presence of snow cover insulating the medium or not), while resonance frequency starts to increase ; 3) deep freezing phase (late winter), when temperature is stabilized due to insulation by permanent snow cover, while the freezing front propagates deeper, increasing the resonance frequency ; 4) Shallow thawing phase (early spring), when temperature reaches 0°C and stay during a zero-curtain period, indicating phase change together with melting water percolating into the active layer and sometimes re-freezing, while resonance frequency drops due to thawing of surface layers ; 5) deep thawing phase (late spring and early summer), when the heat wave propagates deeper in the medium, keeping the decrease in resonance frequency up.






In comparison with other passive seismic methods, as relative seismic velocity variations computed from ambient noise correlation that has already been applied in Gugla (Guillemot et al., 2020), the spectral analysis of seismic noise (presented here) is easier to process. Combined with the modal analysis of a mechanical model of the site, the spectral content accurately records the seasonal freeze-thawing cycle, reinforcing observations from ambient noise correlation (Guillemot et

al., 2020). Beyond these similarities, the main difference between these two methods is their depth sensitivity. Frequency resonance focuses on isolated frequencies, whereas ambient noise correlations exploits the whole spectrum, thereby surveying a larger range of depths. To quantify this difference between the two methods, we computed sensitivity kernels for each one. It consists in evaluating the changes (of frequency or dV/V) after a 50% increase of seismic velocities $V_p$ and $V_s$ for a 0.5 m thick sublayer along the depth of the modeled rock glacier. All the parameters are those of the summer models (for

Gugla in Figure 9, for Laurichard C00 and C05 in Figure 8(c)), and kernels have been computed for all these three sites. These results are presented in Fig. 15: 1) for ambient noise correlation method: the theoretical relative velocity change of the Rayleigh wave ($dV/V$) is computed by dispersion curve difference using the Geopsy package (Wathelet et al., 2004) ; 2) for modal analysis method, the resonance frequency of the fundamental mode of the vibrating structure modeling the rock glacier is obtained using Comsol software. For both methods, their kernels have been normalized by their maximum value

along depth, allowing an estimation of the depth where the sensitivity of the method is the highest. The results are shown in Figure 15 for Laurichard C00 sensor, while other sites are not presented but yield similar results. For all sites, modal analysis is most sensitive at a relatively shallow depth (5 m for Gugla, 4 m for Laurichard C00, 3 m for Laurichard C05) in the active layer, whereas ambient noise correlation has a broader sensitivity, including shallower and deeper layers depending on the frequency band (the lower the frequency, the deeper the penetration). Therefore, the modal analysis permits to easily

evaluate the state of freezing of rock glaciers, surveying mostly the depth range between 2 m and 8 m, including the active layer ($< 5$ m), while ambient noise correlation at low frequencies allows the same monitoring over a broader range of depths but requires additional data processing. Furthermore, ambient noise correlation may provide less stable results at high frequencies (up to 14 Hz, for the Gugla study (Guillemot et al., 2020)), preventing any interpretation of the chaotic results. In this scenario, the two passive seismic methods may be combined to obtain stable results along the whole depth of the rock

glacier. As many other geophysical techniques, the present study is therefore to be considered as one element among other parts of a global monitoring strategy.

## 6 Conclusion

For two rock glaciers, we monitored the resonance frequencies of vibrating modes during several years thanks to seismic noise measurements. These frequencies show seasonal variations, indicating a freeze-thawing effect on elastic properties of

the structure. Assuming vibrating systems, we performed 2D mechanical modeling of rock glaciers, which fit well the recorded resonance frequencies. By quantitatively modeling the increase of rigidity due to freezing in wintertime using a



poroelastic approach and models derived from geophysics, we have reproduced the observed seasonal variations, thus highlighting the sensitivity of resonance frequency on freeze-thawing cycles.

The results of this modal analysis have been obtained from a model constrained by geophysical investigations, as Ground
Penetrating Radar and seismic tomography surveys. This study shows that the two approaches (spectral analysis of seismic data, combined with GPR and seismic refraction) provide a consistent understanding of seasonal variations of rock glacier rigidity, mainly forced by the freezing effect of those porous media.

Among passive seismic methods on rock glaciers, the spectral analysis appears as an easy and effective monitoring tool of the active layer, which is subjected to significant seasonal changes. At greater depths and lower frequencies, the seismic data
can be preferably processed using a pair of stations by computing ambient noise correlation. This can be useful to complement observations of resonance frequencies, in addition to bringing new insights to other deeper processes, such as groundwater or structural changes within rock glaciers. On the long term, seismic vibrations offer the possibility to monitor the effect of global warming on the permafrost degradation.

**7 Acknowledgments**

Seismic data of Laurichard rock glacier are available at the French RESIF seismological portal (http://dx.doi.org/10.15778/RESIF.1N2015). Seismic data of Gugla rock glacier are available upon request at CREALP. Some valuable information about the geophysical campaign, data from boreholes and their interpretation were shared with permission from CREALP (see weblink in references). We are particularly grateful to Benjamin Vial and Mickaël Langlais (ISTerre - SIG), Guillaume Favre-Bulle (CREALP) and Ludwig Haas (Wallis canton) and the geological department of
Wallis for their invaluable assistance with fieldwork, site maintenance and seismic data retrieval of Gugla rock glacier. This work is supported by the OSUG@2020 Labex, the VOR-UGA program, the CNRS-INSU program, and the ANR LabCom GEO3iLab. For Laurichard site, one part of the research was partially supported by Lautaret Garden-UMS 3370 (Univ. Grenoble Alpes, CNRS, SAJF, 38000 Grenoble, France), member of AnaEE-France (ANR-11_INBS_0001AnaEE_Services, Investissement d'Avenir frame) and of the eLTER-Europe network (Univ. Grenoble Alpes, CNRS, LTSER Zone Atelier
Alpes, 38000 Grenoble, France).

**Appendix A: Frequency picking of earthquakes signals**

In addition to spectral analysis from ambient noise, PSD of earthquakes signals emerging from noise have also been computed. For both sites, such signals have been sorted out from a catalog of earthquakes (magnitude M>2). For the Laurichard area, we used all earthquakes recorded by the Sismalp catalog[2]. We thus applied the same processing than for
noise (without any clipping) for the 60 s-long raw trace containing the signal of earthquakes, and finally track resonance

---

[2] https://sismalp.osug.fr/evenements





frequencies of these quakes by maxima picking (Figure 2). For the Laurichard site, we used seismic traces of another station located in a stable area at Lautaret pass (see Figure 1), named OGSA (RESIF, 1995). Since OGSA is considered as a reference station, we computed a site-to-reference spectral content to evaluate the specific frequency peaks of the Laurichard rock glacier (see Figure 2). In this way, we ensured that those picked frequencies are related to the modal signature of the

rock glacier. Overall, this method of spectral analysis allows comparing the spectral response of the structure to low (seismic noise) and higher (earthquakes) levels of excitation.

The resonance frequencies estimated using earthquake signals (white dots on Figure 17) appear similar to the ones estimated from noise for C00 seismometer. However, there are more discrepancies for sensor C05. For this sensor, the peak frequencies determined from seismic signals show more fluctuations than when picking resonance frequencies from PSD of

seismic noise.

**Appendix B: Modal analysis using finite element method**

The finite-element method aims to numerically estimate the resonance frequencies of a vibrating structure, by solving the Newton's second law for the displacement of the considered degrees of freedom $V(t)$ (Bathe, 2006). Assuming free-equilibrium and no attenuation, the equation is:


$$[M]\{\ddot{V}(t)\} + [K]\{V(t)\} = \{0\} \tag{1}$$

where $[M]$ is the global mass, $[K]$ is the global stiffness matrix, and the dot means time derivative. Both $[M]$ and $[K]$ matrices are obtained by correctly assembling the respective element matrices, in accordance with finite-element method (Bathe, 2006).

As a result, the solutions of equation (1) have to be of the form

$$\{V(t)\} = \{\psi\} \sin[\omega(t - t_0)] \tag{2}$$

where $\{\psi\}$ refers to a vector of order $n$, $\omega$ is a constant identified to the corresponding pulsation of the vibrating mode $\{\psi\}$, and $t$ and $t_0$ are respectively the time variable and an arbitrary time constant.

Equations (1) and (2) provide the generalized eigen problem:

$$[M^{-1}K]\{\psi_j\} = \omega_j^2\{\psi_j\} \tag{3}$$





By solving this linear system, we can deduce the modal parameters: the $n$ eigenvalues $\omega_j^2$ (with $0 \leq \omega_1^2 \leq \omega_2^2 \leq \cdots \leq \omega_n^2$ ) and the corresponding eigenvectors $\{\psi_j\}$. The eigenvector $\{\psi_j\}$ is called the $j$-th modal shape vector that vibrates at the frequency $f_j = \omega_j/(2\pi)$.

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



Figure 1: (a,c) Aerial views of the two sites (from ©Google Maps and ©Google Earth). (b) Global topographic map of the
western part of the alpine belt, centered around the French and Swiss Alps. (d) Digital elevation model of the Laurichard
rock glacier, and location of the Miniature Temperature Dataloggers (MTD) that monitor ground thermal regime at the sub-
surface, the seismometers, and the geophysical surveys: Ground Penetrating Radar (GPR, red and blue points); seismic
refraction profile (yellow points). The mean annual surface velocity fields (over the period 2012-2017) is revealed by the
background color. The 6 continuous seismometers are marked by large circles (C00 to C05). The dashed red line depicts the
C00-C05 profile used for the bedrock depth estimation (see Figure 8). (e) Digital elevation model of the Gugla rock glacier
(front in red line) and location of instrumentation, the seismic refraction profile and geomorphologic features. The mean
annual surface velocity measured between 2014 and 2015 by photogrammetry (CREALP, 2016) is also shown on this map.



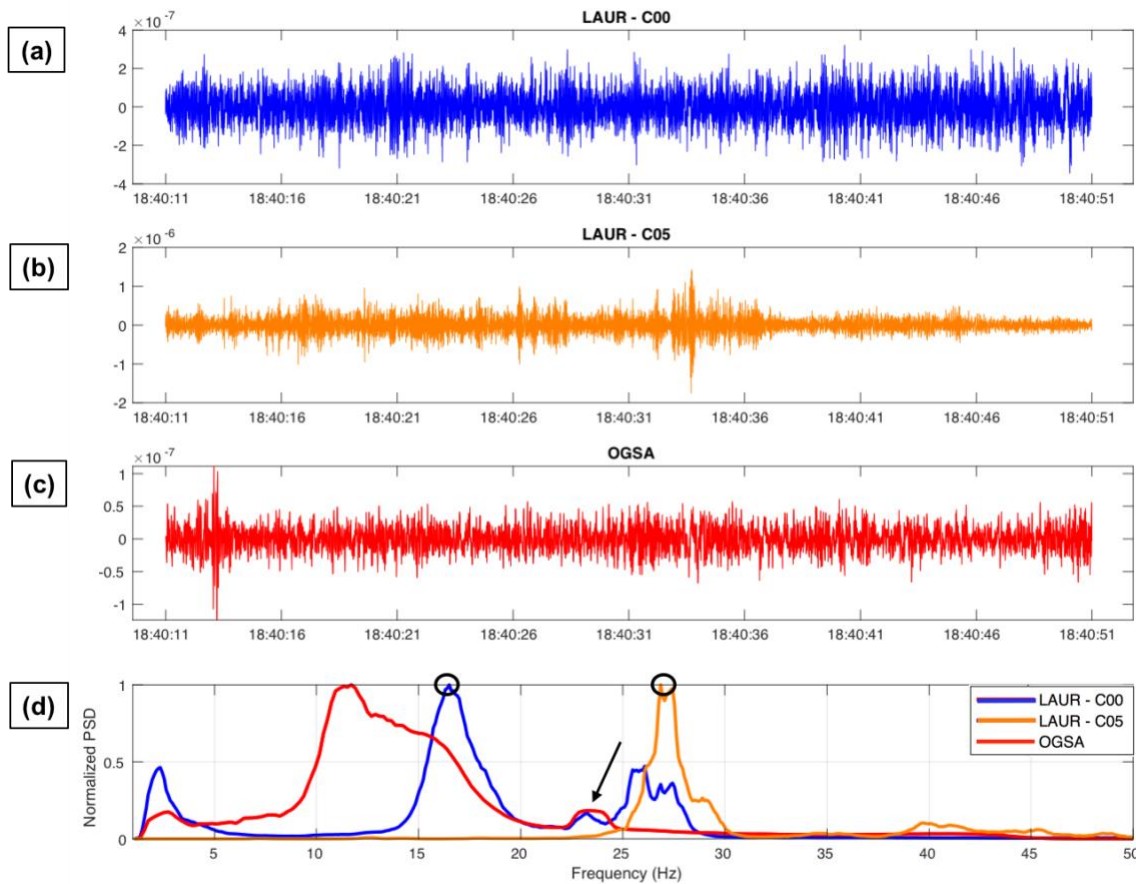

Figure 2: Seismic recordings of ambient noise (vertical ground velocity in m/s, after >1Hz filtering and instrumental deconvolution) recorded by sensors C00 (a) and C05 (b) in Laurichard rock glacier, and by OGSA station at Lautaret pass (c), the 5th April 2019 at 6 PM. (d) The normalized PSD of the respective signals. Black circles highlight the maxima of these spectrograms that have been picked by using our method (details in the text) for sensors on Laurichard rock glacier. The black arrow shows the stable peak at 23 Hz, interpreted as anthropogenic.








Figure 3: Normalized Power Spectra Density (PSD) from hourly spectrograms of the passive seismic recordings of Laurichard site, respectively from (a) C00 seismometer and from (b) C05 seismometer. The bold black line denotes the moving window average of hourly spectrogram maxima. Snow cover and melting periods are both figured by white and blue boxes above, respectively.



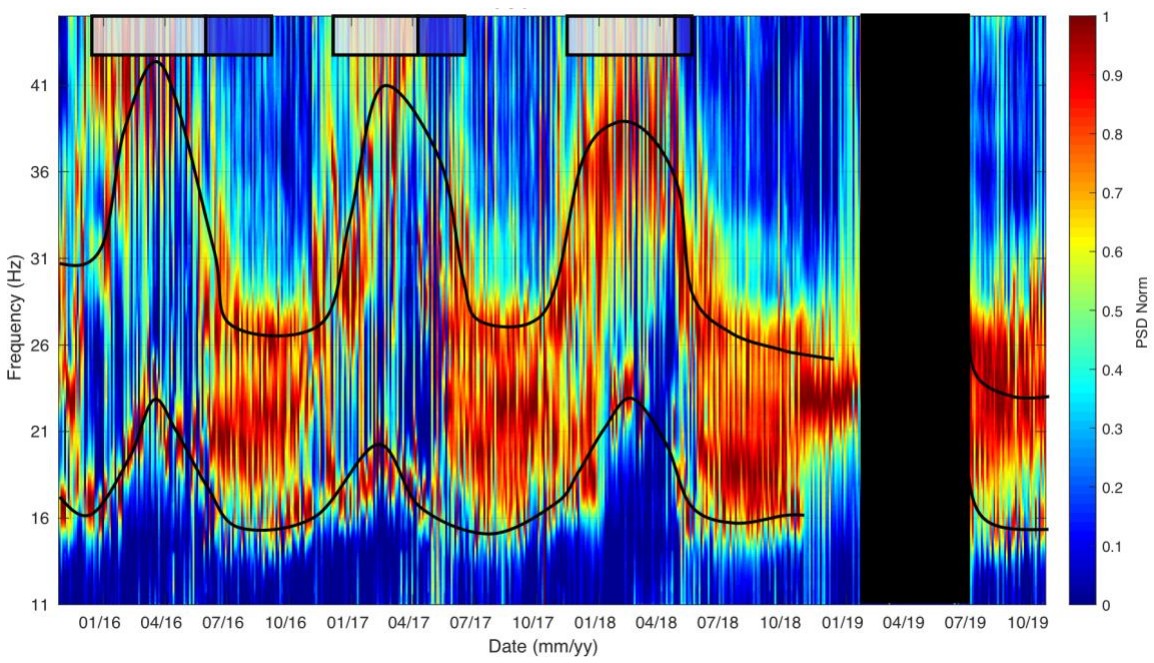


Figure 4 : Normalized Power Spectral Density (PSD) from hourly spectrograms of the ambient noise recordings of Gugla rock glacier, from C2 seismometer. Note the two bold black lines that roughly highlight the two picked spectral modes, for visibility purposes. Snow cover and melting periods are both figured by white and blue boxes above, respectively. No-data period is marked by a black box.


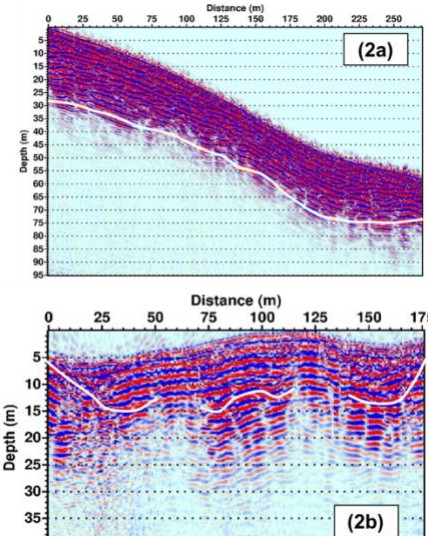

Figure 5: GPR results for Laurichard rock glacier, with: (1a) Common Middle Point GPR data acquired with a 100 MHz unshielded antenna. (1b) velocity analysis displaying the semblance according to apparent velocity and propagation time. The red curve indicates the picked maximum of semblance. (2) Common-offset 25 MHz profiles : (a) Longitudinal profile.

Elevation corrections have been divided by a factor of 2 for visibility purposes. (b) Transverse profile. On both cases, the vertical/horizontal ratio axis has been scaled by a factor of 2.4, and the bedrock interface is highlighted by a white curve.

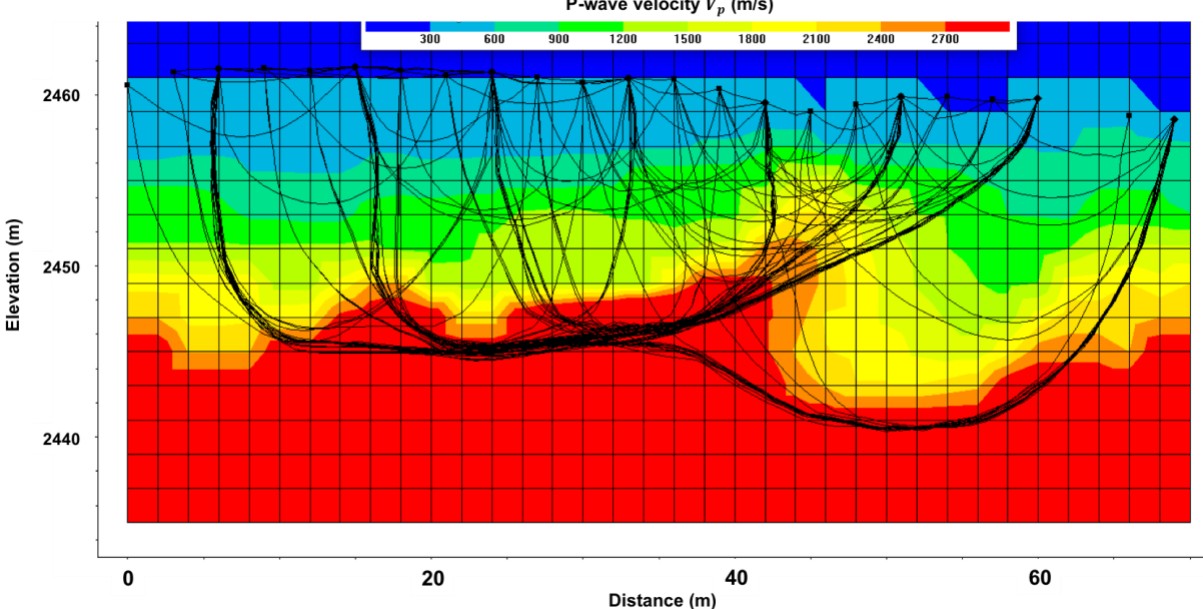

Figure 6: P-wave velocity distribution obtained from a seismic tomography acquired along the transversal profile of seismic refraction (yellow line in Fig. 1a) in summertime. The different ray paths are shown with black curves. The seismic

velocities were used to constrain bedrock depth and P-wave velocity profiles for the mechanical modeling.

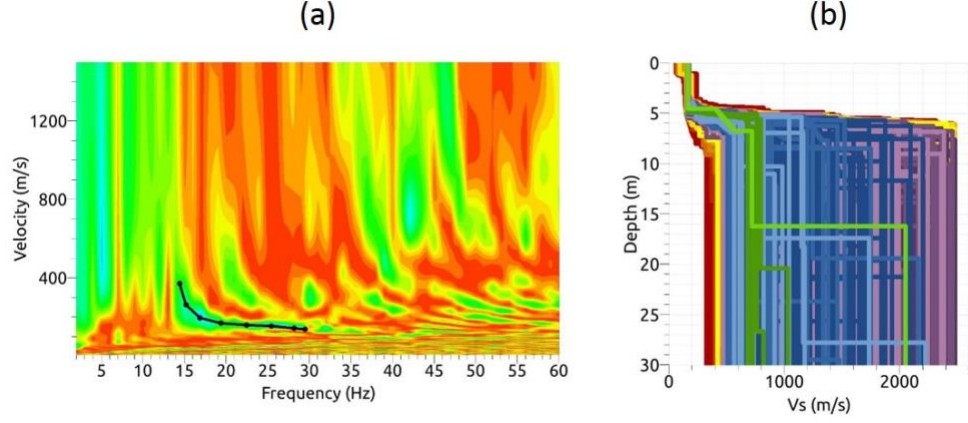





Figure 7: Multichannel analysis of Rayleigh waves propagating within the rock glacier. (a) semblance velocity-frequency map highlighting several modes, the fundamental dispersion curve being picked as indicated by the black line and (b) Vs distribution versus depth derived from the inversion of the fundamental dispersion curve. Colors indicate the RMS error
between synthetic and picked fundamendal dispersion curves (best fitting model in green).

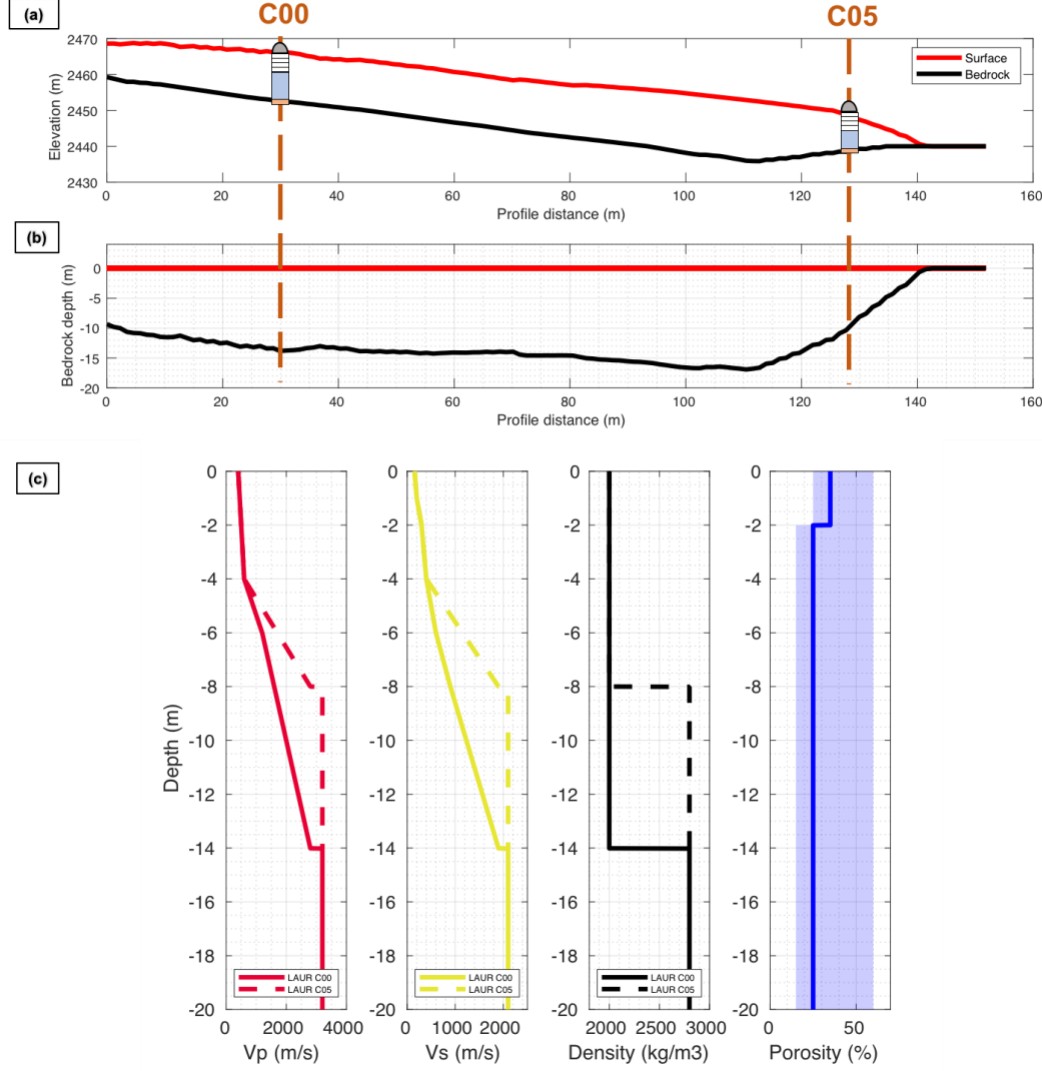

Figure 8 : (a) Cross-section of the Laurichard rock glacier Digital Elevation Models (DEM) of the surface and of the bed (taken for the bedrock) along the C05-C00 line. (b) The same profile, with the DEM of the surface as the reference. The vertical axis is then the bedrock depth starting from the surface. For both figures the location of the seismometers is
indicated. (c) Seismic velocity models of the Laurichard rock glacier (continuous line for the C00 case, dashed line for the C05 case), based on geophysical investigations (seismic refraction) and the bedrock depth estimation, determined from DEM difference (see 1b). The only difference between the two cases is the bedrock depth, and consequently the seismic velocity





gradient of the permafrost layer. On right panels the density and the medium porosity profile are shown (solid blue curve), with low and high limits of the porosity used in the following mechanical modelling.

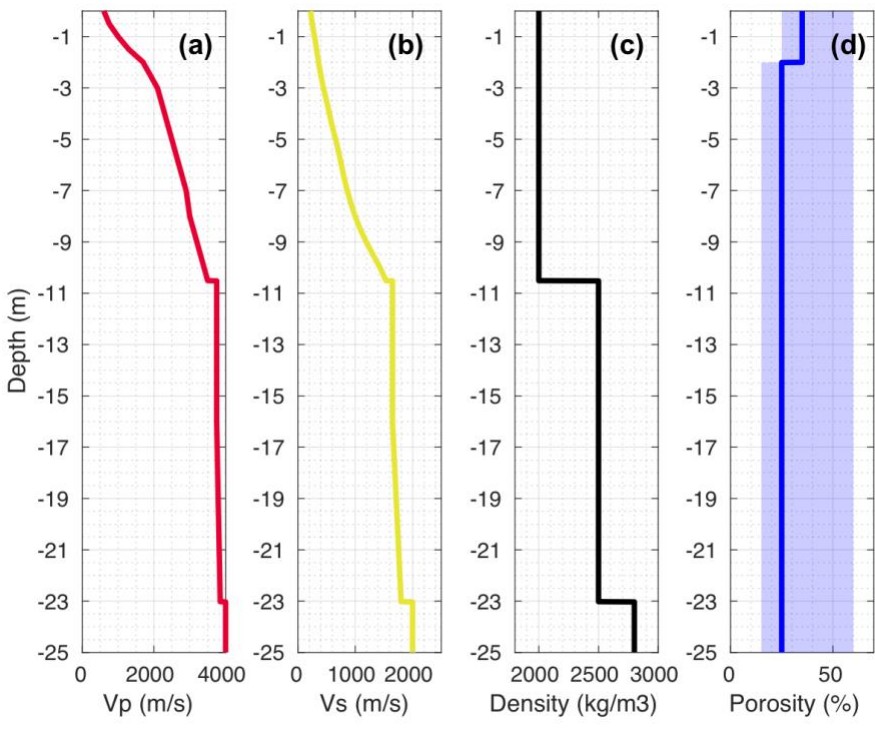


Figure 9 : Seismic velocity models of the Gugla rock glacier, based on geophysical investigations (seismic refraction) and from borehole data, with P-wave velocity (a) and S-wave velocity (b) profiles. The density (c) and the medium porosity ((d), solid blue curve) profiles are also shown, with low and high limits of the porosity used in the following mechanical modelling.





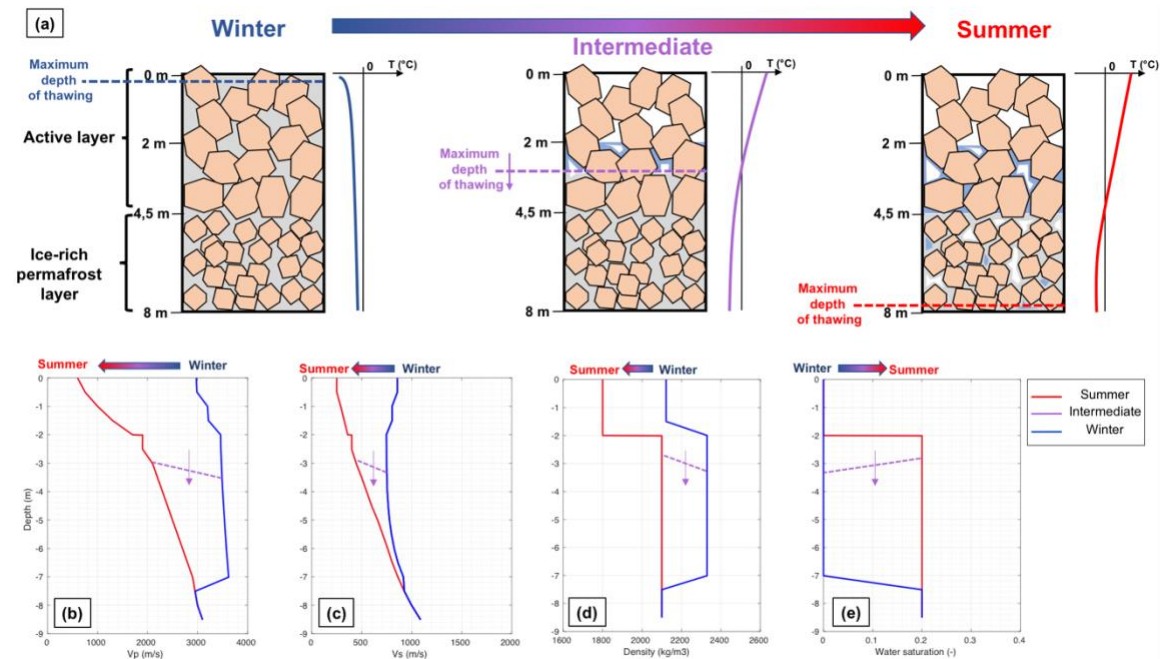


Figure 10: (a) Schematic cross-section of the Gugla rock glacier in winter (left) and in summer (right), and during the transition between them (intermediate state of thawing, in middle), as well as a schematic temperature profile associed with each of them, showing the main assumptions of the freezing modeling methodology by a poroelastic approach described in the text. The porous medium is composed of a rock matrix (in orange) and pores filled by water (in blue) or ice (in grey).

With respect to the maximum depth of thawing varying from the surface to the ZAA depth, the evolution of parameters used by the model is respectively showed : P-wave velocity (b), S-wave velocity (c), the averaged density (d) and the water saturation (d). The values in summer are obtained from geophysics and boreholes, whereas the values corresponding to a frozen state (pores fully filled by ice, no more liquid water) are obtained by the 3-phases poroelastic model.





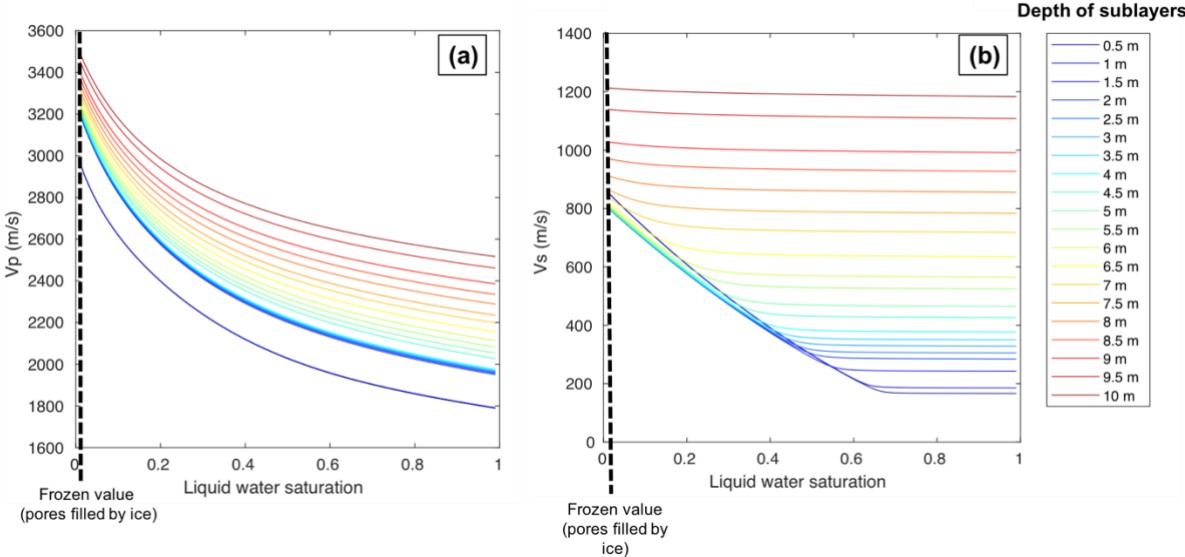

Figure 11: (a) Evolution of P-wave velocity $V_p$ with respect to the ratio between water and ice filling the pores, resulting from the Biot-Gassmann type three-phase poroelastic model, applied to the Laurichard rock glacier (for C00 seismometer). The different curves correspond to the different sublayers, whose depth is indicated in the right panel. The $V_p$ values used to model the winter state are those for liquid water saturation tending towards zero. (b) Same results for S-wave velocity $V_s$.





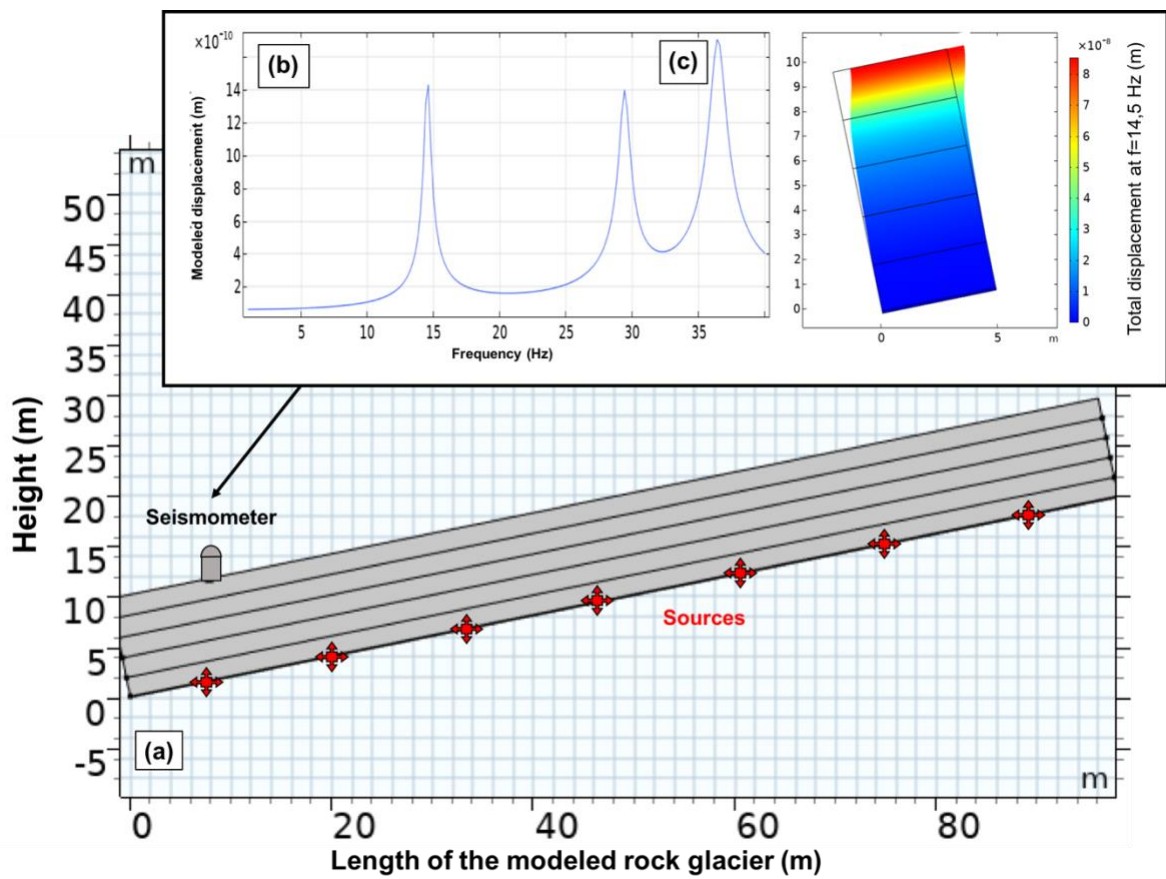

Figure 12: (a) First 2-D model of the terminal part of Laurichard rock glacier, with realistic longitudinal dimension (length) and location of synthetic seismic sources and seismometer. (b) Output of the Frequency Response Function (FRF) of this model, as the vertical displacement recorded by the seismometer along frequencies. Peaks of this curve indicates resonance frequencies of modes of the modeled structure with a vertical component. (c) The 2-D rock glacier model with reduced longitudinal dimension (5 m), and with symmetrical conditions at boundaries perpendicular to the slope. Since the FRF of this model shows the same peaks than the first one, this reduced model is only used for the modal analysis for the sake of simplicity.





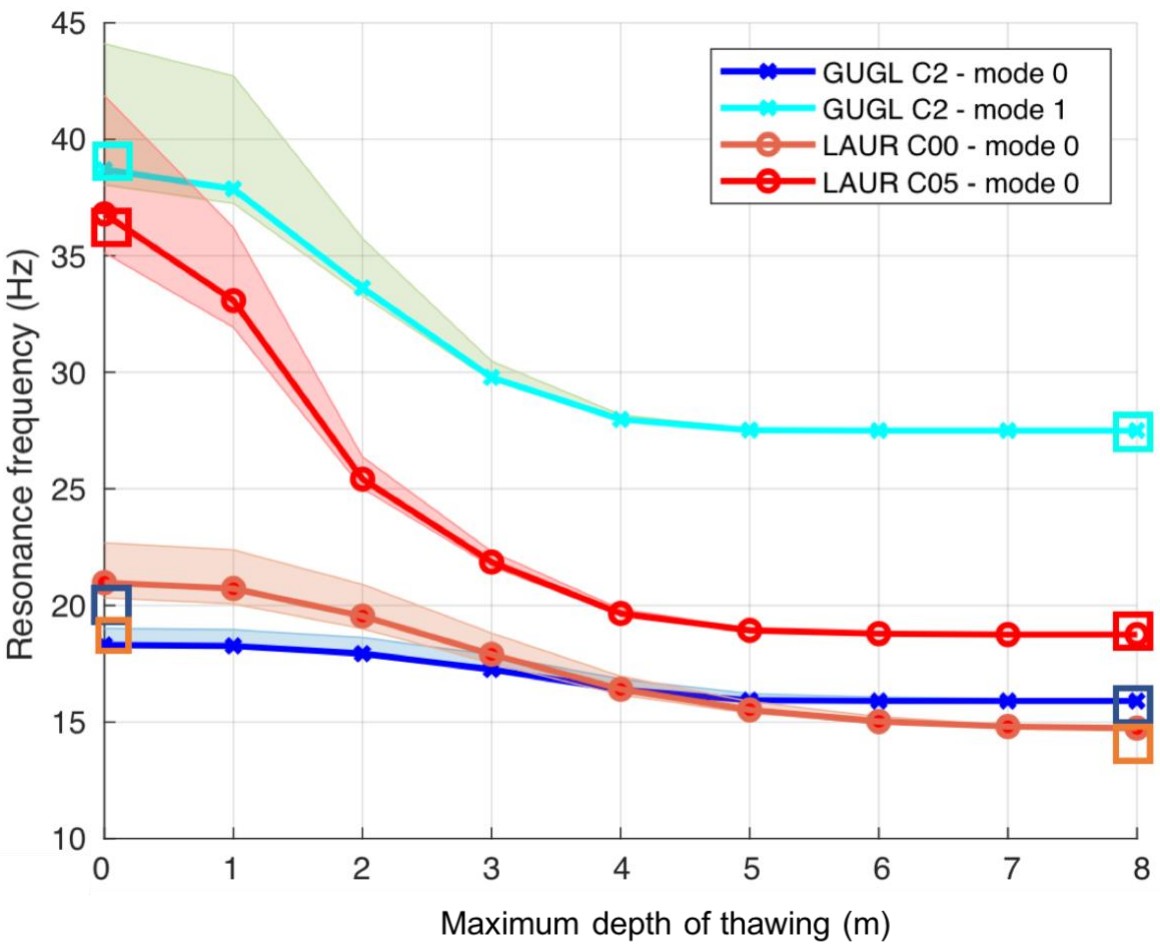

Figure 13: Results of the modal analysis for Gugla (GUGL) and Laurichard (LAUR) rock glaciers. The evolution of the resonance frequency of the respective synthetic modes is plotted, according to the maximal depth of thawing from the surface to the ZAA (8 m depth): left stands for the state of maximal freezing in winter (frozen medium until 8 m depth), right for the summer (unfrozen medium). The range of measured resonance frequency values is shown by the squares in the corresponding colors (estimated from Figure 3 and Figure 4 for respective sites). Errorbars (in transparence) show the influence of porosity on the results : the low (high) limit of errorbars shows the results for extremely low (extremely high) porosity profile.

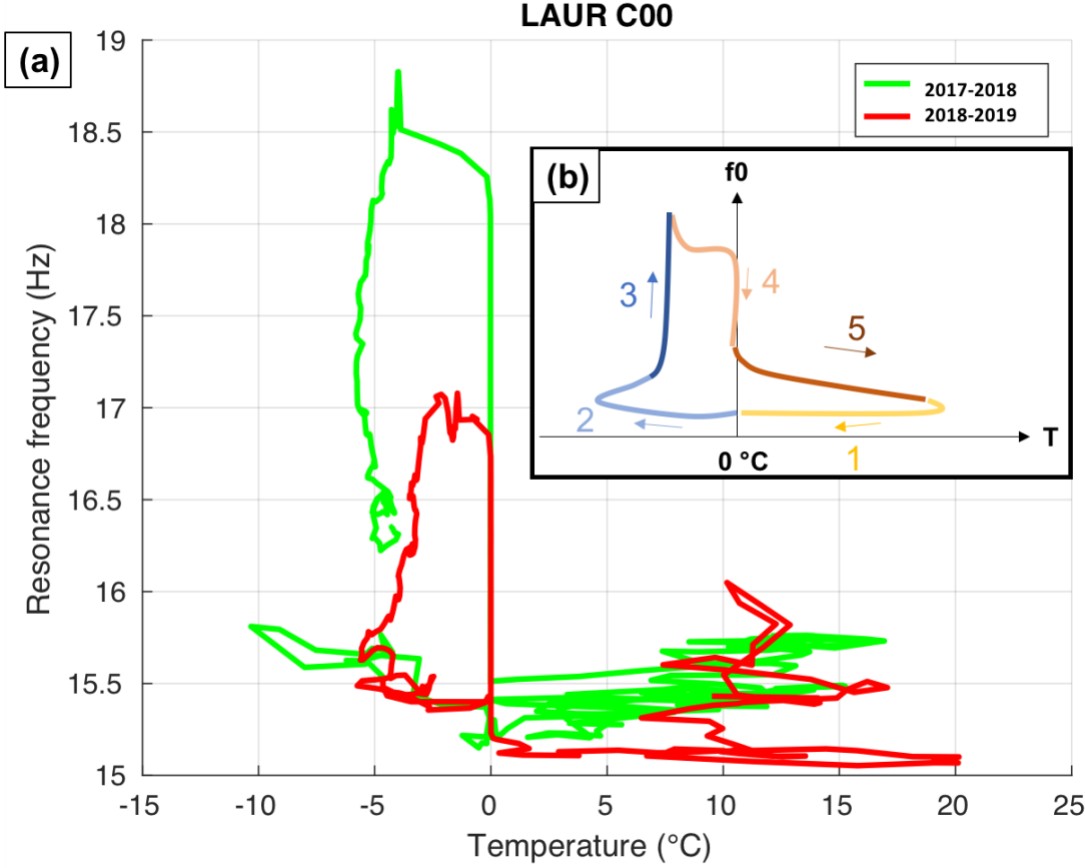


Figure 14: (a) Diagram of daily-averaged ground temperature (recorded by Miniature Temperature Dataloggers) versus daily-averaged resonance frequency of the first mode recorded by C00 seismometer at Laurichard rock glacier. The green curve corresponds to data from December 2017 to November 2018, while the red curve corresponds to data from December 2018 to October 2019. (b) Schematic generalization of the ground surface temperature dependency of resonance frequency with freezing-thawing cycle, showing an hysteretic loop composed of five phases described in the text.





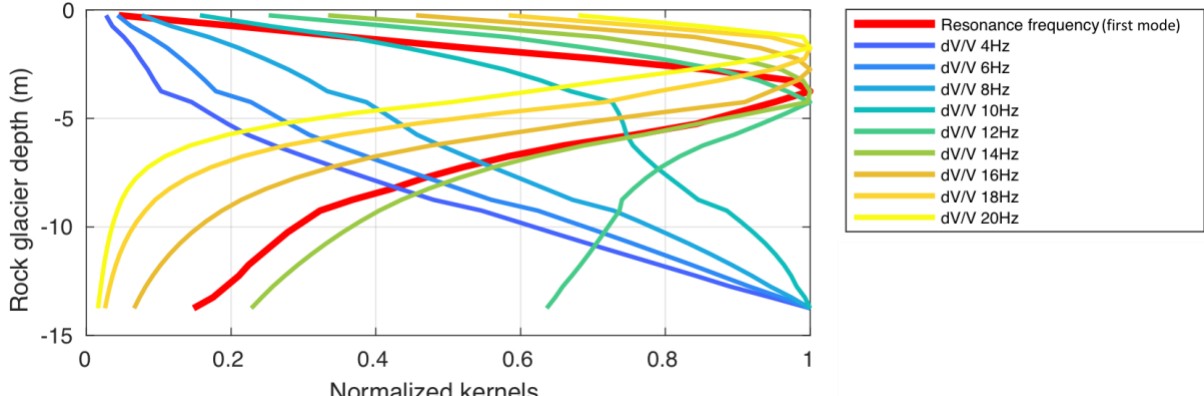

Figure 15 : Example of (normalized) depth sensitivity kernels of the two passive seismic methods for the Laurichard (C00 seismometer) rock glacier model. The red curve corresponds to the modal analysis (resonance frequency of the first vibrating mode). The other curves correspond to the ambient noise correlation (relative change of the Rayleigh wave velocity $dV/V$,

depending of the frequency, shown by the other colours). At high frequencies ($> 14$Hz), $dV/V$ is most sensitive at shallower depths than the resonance frequency of the first mode, whereas at low frequencies ($< 14$Hz), $dV/V$ is most sensitive at deeper depths than resonance frequency.





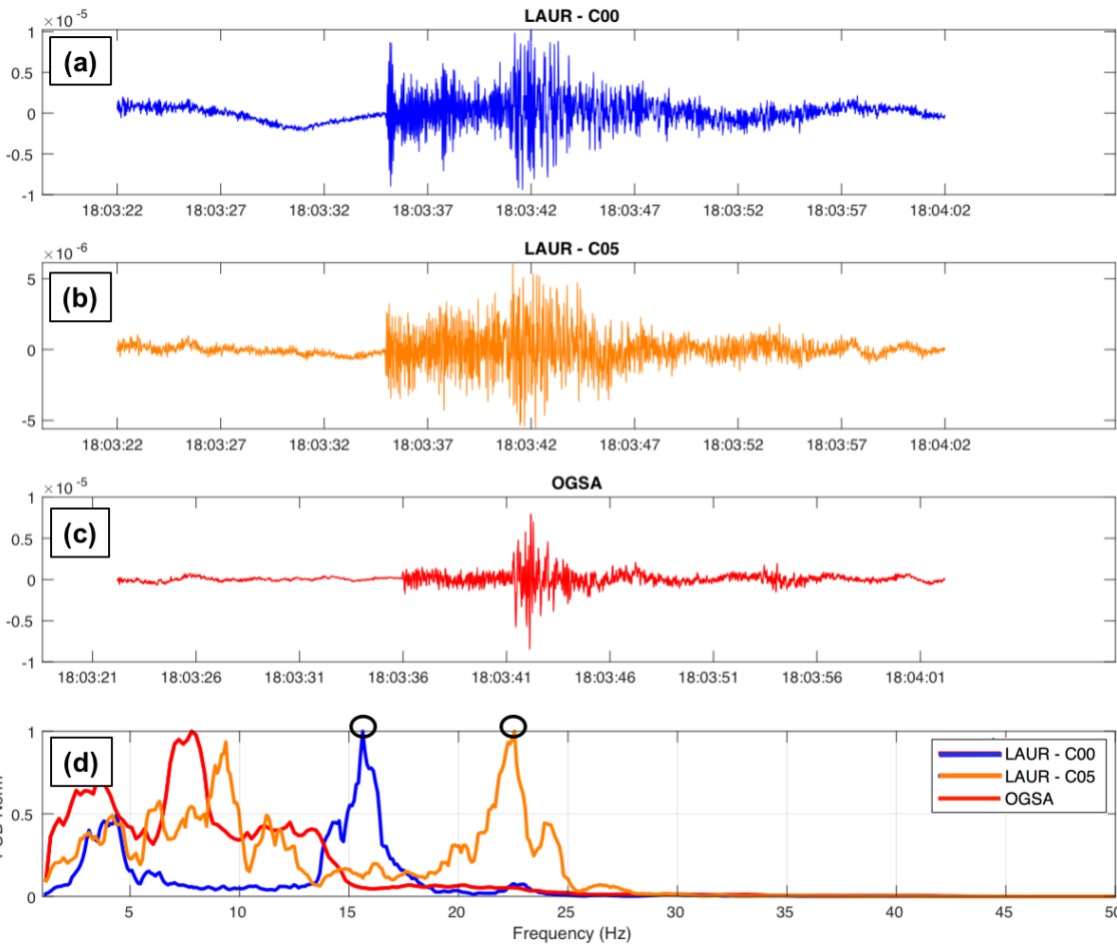

Figure 16: Seismic signals of an earthquake (vertical ground velocity in m/s) recorded by sensors C00 (a) and C05 (b) in
Laurichard rock glacier, and by OGSA station at Lautaret pass (c), the 29rd June 2018 at 6 PM. (d) The normalized PSD of
the respective signals. Black circles highlight the maxima of these spectrograms that have been picked by using our method
(details in the text). The same method has been used for ambient noise recordings.



Figure 17: Normalized Power Spectra Density (PSD) from hourly spectrograms of the passive seismic recordings of
Laurichard site, respectively from (a) C00 seismometer and from (b) C05 seismometer. The bold black line denotes the
moving window average of hourly spectrogram maxima. For each recorded earthquake (M>2), the local maxima have been
automatically picked (white dots) if it appears significantly on the spectrogram. Snow cover and melting periods are both
figured by white and blue boxes above, respectively.