# Peer review of "Modal sensitivity of rock glaciers to elastic changes from spectral seismic noise monitoring and modeling"

_The Cryosphere, 2020_

## Referee Comment (RC1) · Andreas Köhler (Referee) · 31 Aug 2020

**Review of manuscript "Modal sensitivity of rock glaciers to elastic changes from spectral seismic noise monitoring and modeling"**

This is a very good and comprehensive study demonstrating the ability of passive seismic measurements for time-lapse monitoring of near-surface structures in combination with other geophysical methods which provide more high-resolution, but time-restricted information. The presented use of modal analysis in particular is a novel and promising approach for rock glacier monitoring. The results presented in this study support the capability if the method.

(1) I would be interested in a brief discussion on how the HVSR (Horizontal-to-Vertical Spectral Ratio) method is related to the modal analysis of single component seismic data. In case of HVSR, the argument often used is that normalizing the horizontal spectrum by the vertical will reduce the source effects and thus enhance the resonance spectrum of the site. HVSR peaks are interpreted either as SH wave resonance (see equation on Page 6, line 190) or maxima of the Rayleigh wave ellipticity in a layered medium. Would it make sense to present the results of this study as HVSR time-lapse spectra instead of single component spectra (assuming the three-component seismometer have been used)? I would expect to see the same temporal variability by potentially reducing at the same time some spectral peaks related to local sources. Is the 3D nature of the rock glacier the reason for not using spectral ratios?

(2) The ability of the HVSR method for time-lapse permafrost monitoring has been recently investigated in a few studies:

Abbott, R., Knox, H. A., James, S., Lee, R., and Cole, C.: Permafrost Active Layer Seismic Interferometry Experiment (PALSIE), Tech. rep., Sandia National Laboratories (SNL-NM), Albuquerque, NM (United States), available at: https://prod.sandia.gov/techlib/access-control.cgi/2016/160167.pdf (last access: 7 January 2019), 2016.

Kula, D., Olszewska, D., Dobiński, W., and Glazer, M.: Horizontal-to-vertical spectral ratio variability in the presence of permafrost, Geophys. J. Int., 214, 219–231, https://doi.org/10.1093/gji/ggy118, 2018.

Köhler, A. and Weidle, C.: Potentials and pitfalls of permafrost active layer monitoring using the HVSR method: a case study in Svalbard, Earth Surf. Dynam., 7, 1–16, https://doi.org/10.5194/esurf-7-1-2019 , 2019.

It might be useful to have a look at these studies. Please feel free to not cite them if they are not relevant for the current work (especially since a paper of mine is included).

(3) I miss a figure directly comparing the temporal evolution of the measured resonance frequencies with the modeled once for different states of thawing. Furthermore, a figure comparing the measured and modeled amplitudes spectrum for a particular time would be very useful (for example overlaying figure 2d). In my opinion such figures are more important for demonstrating the reliability of the method than showing the GPR and active seismic results in Fig 4,5 and 6. Those can be moved to the Appendix.

Minor comments:

General: Please check if all commas are needed.

The first sentence of the abstract and the Introduction are identical. Please consider rephrasing.

Line 42: highly challenging?

Line 49: Do you mean "coast-intensive"? Or "…remain cost-effective only when limited to one single …"?

Line 61: active permafrost layer

Line 73: Please explain "as a bending beam"

Line 76: "Our goal … of the rock glacier and the time variability of their resonance frequencies which gives hints …"

Line 78-79: "… are numerically modeled …"

Line 81: "In the second part, …"

Line 97: Something is wrong with formatting of exponents

Line 123: one bracket too much

Line 130: Are these three-component sensors? Was maintenance required during the measurements (releveling etc.)?

Line 140: I think the time-lapse resonance spectra of the other sensors should be included as supporting materials or in the appendix (do not need to be discussed)

Line 167: "Continuous seismic monitoring systems are composed …" (?)

Line 171: "… between several sensors and to monitor …"

Line 195: I suppose the vertical component is used?

Line 200: This is unclear. Please rephrase to describe how peaks are picked automatically.

Line 261: Just to avoid misunderstanding: Is the modelling of resonance frequencies done in full 3D, 2D or for a particular location and 1D model below?

Line 267: "cost-effective" See above.

Lines 376, 409, 449, and 452: paragraph/section numbers wrong

Line 476: "inter-annual climate variability" Is "climate" the right word here? I guess the constant climate at a particular site includes the inter-annual variability of temperatures.

Line 491: "observed gap" Do you mean "observed difference"?

Line 496: "in combination with"?

Line 498: Remove "Furthermore"

Line 515: Sentence "Frequency resonance focuses on …": I am not sure if this is correctly formulated. The resonance frequency in general is also an effect of seismic waves propagating through the whole structure. It just depends on the considered frequency band. Here, I agree with your conclusion that resonance works well at high frequencies (and thus for shallower depths) where most of the changes occur, while noise correlation does not work so well due to lack of correlation at high frequencies if the inter-station distance is too large. So, the different sensitivities are mainly because of the nature of the ambient noise wavefield and the sensor network set-up, not because of the depth sensitivity of both methods as such. See for example the study of James et al, where noise correlation could be used to measure very shallow variability with closely located sensors.

Line 542: geophysical measurements

Appendix A: I am not sure if it necessary to include the results of earthquakes since the results are not much discussed. If they are included, one would like to know where the discrepancy compared to noise comes from.

Figure 2: Showing the noise waveform record is not necessary in my opinion. Instead, please also add a plot like (d) for Gugla.

---

## Referee Comment (RC2) · Anonymous Referee #2 · 25 Sep 2020

**Review of tc-2020-195 "Modal sensitivity of rock glaciers to elastic changes from spectral seismic noise monitoring and modeling » by Guillemot et al.**

The study presents a new methodology that uses the resonant frequency of rock glaciers extracted from continuous ambient noise records to monitor the seasonal and interannual changes in the structure of the vibrating glacier. The seismic modal analysis is coupled to a poroelastic model of a 2D porous medium representing the rock glacier, supported by other geophysical measurements. The results indicate the thawing-freezing cycle effects on the resonance of the glacier. This comprehensive study highlights well how the structural changes of rock glaciers can be tracked and monitored with passive seismology, when other geophysical methods are time-consuming and hardly repeatable at such time resolution.

**Major comments:**

(1) I miss somewhere an explanation of the origin of the ambient seismic noise (at frequency > 1Hz). Many studies agree to say that it takes its origin in fluvial processes when the water flow from ice melting in spring/summer creates transient forces on the Earth at the glacier base or on the surrounding ice in englacial channels. The noise recorded in winter may come from other sources in the area. Many studies on ambient noise suggest that a change in the noise sources could lead to a change in the noise spectrum. This renders monitoring studies difficult, especially for ambient noise correlation method. Here you use a modal analysis which should be less sensitive to noise source changes. But still, there remains an open question: are the seasonal variations of the resonant frequencies influenced by changes in the noise or actual structural changes ? I am quite concerned about the abrupt resonant frequency shift you observe at the onset of the melting.
Personally, I am absolutely confident in the interpretation of actual changes in elastic properties of the structure underneath the sensor. However, this question should be raised and discussed.

(2) As a proof of concept and to approve the interpretation of the results, I suggest to try to actually reproduce the seasonal variations of the resonant frequencies with the poroelastic model (as stated in the conclusion Line 542 but wrong). This would strengthen the discussion and the capability of the method. The interpretation is finally based on Fig 14 which represents measured resonant frequencies as a function of the temperature of the ground surface, while this temperature does not tell the whole story on what is happening at depth. So having a hint on the evolution of the active layer and the rigidity of the structure thanks to the best-fitting model output would definitely strengthen your conclusions. In addition, this figure would be very great to track the interannual changes. Finally, I miss a figure showing the temperature time-series (in this additional « proof of concept » figure, or in Fig 3 for example). This would ease the reading of Fig 14 and also highlight melting seasons versus winter.

(3) Sometimes, I think that the structure of the sentences is not smooth enough and lead to confusing statements in English. I have listed some in the comments below which need to be reformulate. In general you should try to keep the sentences short.

**Other comments:**

L48: in depth -> at depth ?

L57-59: References are needed here for both methods (ambient noise vs microseismicity). I understand this is the output of the study by Guillemot et al 2020 as detailed in the following sentence, but other studies have also proven this in glaciated environment.

L89: I suggest to specify « France » after the « Laurichard catchment »

Figure 1d: Please indicate the flow direction with an arrow.

Section 2.1.3: How were the seismometers maintained (leveling, orientation) ? Are they three-component sensors ?
You should also specify somewhere here the glacier thickness beneath the stations (ice thickness was also not specified in the previous section)

L137: « The long periods » -> the longest periods ?

L139: What is the distance between these two stations ?

L140: It could be interesting to have the results for the seismic measurements at the other stations in the appendix.

L144: I suggest to specify « Switzerland » after « Wallis Alps »

L157: Please specify here if the geophones are one or 3 components. Are they deployed directly on the ice ? Were they maintained ?

L169-171: This is the right place to be more specific for glacier microseismicity and  glaciological applications (location of crevasses, basal interface and asperities, water channels) with appropriate referencing.
For ambient noise studies on glaciers, you could refer to the studies of Preiswerk and Walter 2018, Sergeant et al 2020 (for the imaging part) and Lindner et al 2018 (for the monitoring part)

Preiswerk, L. E., & Walter, F. (2018). High–Frequency (> 2 Hz) Ambient Seismic Noise on High–Melt Glaciers: Green's Function Estimation and Source Characterization. *Journal of Geophysical Research: Earth Surface*, *123*(8), 1667-1681.
Sergeant, Amandine, et al. "On the Green's function emergence from interferometry of seismic wave fields generated in high-melt glaciers: implications for passive imaging and monitoring." *The Cryosphere* 14.3 (2020): 1139-1171.
Lindner, F., Weemstra, C., Walter, F., & Hadziioannou, C. (2018). Towards monitoring the englacial fracture state using virtual-reflector seismology. *Geophysical Journal International*, *214*(2), 825-844.

L171: add « to » before monitor.

L174: « on it » -> « on site »

L176: « tracking dynamic parameters » -> « tracking the evolution of »

L180: « The seismic noise averaging properties » What does this refer to ?

L182: « The PSD is simply defined by averaging the intensity of the FFT » You average it by what ? The (inverse of) time of integration ? I do not see any averaging in the proposed equation.

L185. Preiswerk et al 2019 also investigated the resonant glaciers with different geometries which imply 1D, 2D and 3D resonances through HVSR, time-frequency dependent polarization and modal analysis. This study should be cited, here maybe or elsewhere.

Preiswerk, L. E., Michel, C., Walter, F., & Fäh, D. (2019). Effects of geometry on the seismic wavefield of Alpine glaciers. *Annals of Glaciology*, *60*(79), 112-124.

L191: « in absence of geometrical changes » maybe specify here « (i.e. ablation/accumulation) »

L191: missing comma after Young's modulus.

L195: Which component ?

L200-204: The picking method is not clearly described. Additionally, please specify why you expect to have frequency peaks above 10 Hz on the glacier ? Is it related to the glacier morphology and more specifically to the ice thickness ?

Figure 2d: It seems that you still see the 23 Hz anthropogenic peak at station C05 (as expected) but it is not obvious because of the normalization. Maybe indicate this in the caption.

L212: « The results of PSD » -> « The spectrograms of PSDs »

L218: « no subglacial resonating water-filled cavities was known on the site ». I suspect you refer to moulins which extend from the glacier surface to (probably) the glacier base (this is what it sounds as you cite Röösli et al 2016) ? If yes, the part of the sentence Line 2018 should me modified to point out to moulins. If no, how do you know that there is absolutely no channels in the glacier ?

L225: « resonating structure of the sensor » -> « beneath the sensor » ? Do you refer to the glacier or the shelter around the sensor ?

Section 3.2: see major comment: I miss somewhere an explanation of the origin of the ambient seismic noise (at frequency > 1Hz). Many studies agree to say that it takes its origin in fluvial processes when the water flow from ice melting in spring/summer creates transient forces on the Earth at the glacier base or on the surrounding ice in englacial channels.
You say later (Line 224) that water filling of the resonating structure could be responsible for changes in PSD. Then you say Line 229 that no water was present in the sensor settlement. In fact, the presence of meltwater inside the glacier does affect the seismic noise with strong noise generated in spring/summer and little englacial noise in winter. The noise recorded in winter may come from other sources in the area. This should appear on non-normalized spectrograms. In this case, there remains an open question on the cause of the resonating frequency shift from winter to summer. Is it related to source effects or structural changes ? I am confident in the interpretation of actual changes in elastic properties of the structure underneath the sensor. However, this question should be raised at the end of this section or in the discussion.

Figures 3/4: The blue boxes indicating the melting season are present only for spring. On what observations is it based ? Generally in Alpine glaciers, the melting seasons spread from spring (April/May) to the end of summer (september/october).This would also be consistent with the seasonal cycle you observe for the seismic dominant frequency.
Also related to this comment, I would say in Line 233: « a sudden drop of frequency occurs at the time when melting processes [start to occur in spring and stay stable to lower frequency over the course of the summer]. »

Figures 3-4: I would here display the time series of air or ground temperature + periods of snow cover. This would ease the interpretation in the discussion and Fig 14

Figure 4: Nice observation !

Line 247: I would add « at [spring and] summer time »

Line 261: Do you have a reference for the software ?

Figures 5-7: I suggest to move these figures to the appendix section as I think that the study should be focused on the modal analys/modeling methodology.

Line 391: « appling » -> applying

Line 446: « as shown in Fig 13 for Laurichard and not presented here but similar for Gugla » Results for both glaciers are presented in Fig 13, so you should revise this sentence.

Line 449: Section number missing (and elsewhere)

Line 450: « and compare them to the maximum values of the observed ones ». You should indicate here that data are indicated by squares and rephrase this as the text alone is confusing. Say that you represent here the maximum and minimum bounds of the resonance frequencies you measure for each mode, for the two years - if I understand correctly.

Line 452-453: « Resonance frequencies of these modes match the freq band of measurements below 50 Hz, and generally decrease with thawing ». There is no relation between the two parts of this sentence so you should split it in two. The first part is a bit redundant with the previous sentence of the text. In general I think that section 4.5 could be better rewritten and reorganized with clearer description of the glacier seismic behavior.

Line 454: I would add at the end after « for all cases » « i.e. Gugla and Laurichard »

Line 455: You should remind here that C05 is towards the glacier tongue (on thinner ice H=8m according to your velocity model) and C00 is more upstream (H=14 m).

Figure 13: You could add arrows: one which points toward higher depths of thawing indicating summer, one which points toward 0 indicating winter

Line 465: I would replace « melting periods » by « at the onset of the melting period in spring » (summer is also a melting period).

Line 471: « observed seasonal freq variations » -> observed freq seasonal variations

Lines 487-488: « in 2019 … frequency is lower than is 2018 » Give values !!

Line 488: « In addition to an earlier snow cover period in 2019 than 2018 … » I would reformulate. Did snow falls started earlier in 2019 or was the period for snow cover shifted earlier in time ?
« frozen » -> « refrozen » ?

Lines 475-489: I think this paragraph could be improved with some reorganization. You should highlight your conclusion which is that, on average, resonance frequencies are more sensitive to the intensity of internal thawing (which is influenced by …) than to the air temperature as …
Also you use both the present and the past to describe the observations. You should homogenize this.
Could you explain those differences with your model ? Can different level of porosity and depth of thawing mimic different intensity of thawing ? It would be very great to have a figure showing the time series of the observed resonance frequencies with the model results for different states of thawing.

Lines 489-498: I suggest to remove this paragraph from the discussion which is focused on the thawing-freezing cycle.

Line 499: I would remove « non-linear ».

Line 500: « over the year » -> along the year

Line 501: I would remove « dry » as we do not know if you are referring to the air or the rock glacier.

Lines 521-522: « for ambient noise correlation method, the theoretical relative velocity change of the Rayleigh wave is computed by dispersion curve difference using the Geopsy package» -> I would say « for … corr method, we compute the dispersion curves of the Rayleigh wave using the Geopsy software. The theoretical relative velocity changes are computed by measuring the differences of the dispersion curve with the reference one at each frequency. »

Figure 15: For the modal analysis, you say « first mode » while everywhere in the text you refer to the fundamental mode. This could be confusing. You should specify the resonance frequency, and also in the main text.

Line 538: I would say « continuous seismic noise measurements »

Line 539: This is minor but I would say something like « These freq show seasonal variations that are to be related with changes in elastic properties of the structure underneath the recording sensor, due to freeze-thawing effects ».

Line 541: « which fit well the recorded frequencies » I would specify here or next sentence on what the resonance freq depend in the poroelastic model. Basically the take-home message I keep from your analysis is that the freq depend on the maximum depth of thawing in a porous medium.

Line 542: « we have reproduced the observed seasonal variations » This is a strong statement, you did not reproduced the seasonal cycle exactly but only reproduce the maximum freq in winter due to maximum freezing and minimum freq in summer due to maximum thawing. See my major comment.

Line 551: « insight to other deeper processes » -> into other processes at greater depth

---

## Author Comment (AC1) · 23 Oct 2020

Reply to reviews on the manuscript Modal sensitivity of rock glaciers to elastic changes from spectral seismic noise monitoring and modeling

By Antoine Guillemot, Laurent Baillet, Stéphane Garambois, Xavier Bodin, Agnès Helmstetter, Raphaël Mayoraz, Eric Larose

ContactÂă: Antoine GUILLEMOT ISTerre, Université Grenoble Alpes Grenoble (France) antoine.guillemot@univ-grenoble-alpes.fr

Dear reviewer, dear editor,

[Figure]

We would like to thank you for the constructive review following the submission of our manuscript "Modal sensitivity of rock glaciers to elastic changes from spectral seismic noise monitoring and modeling" to The Cryosphere. We took into account all the comments from the two reviewers and editor. One of the main problems raised concerns the lack of references to the HSVR method. Indeed, we didn't specified clearly that our seismometers are single (vertical) component, so that the HSVR method was inapplicable in our case. However, we added some mentions to previous publications on this method successfully applied on permafrost, since the methodology is very similar to spectral analysis presented in this article. Furthermore, you suggested additional figures to demonstrate the feasibility of our method: one showing the temporal evolution of modeled resonance frequencies, and another one showing modeled amplitude of the spectrum for a particular state of freezing. Unfortunately, both of them are difficult to address accurately to our mind, because of the lack of information concerning thermo-mechanical coupling and 3D effects on the rock glaciers. We detail these points further in our response. As suggested by the reviewers, we also modified some sentences and figures in order to improve both the quality of information that we provide and the readability of the publication. We added also some references to publications about seismic monitoring on permafrost and glaciers, as a completed state of art. Please find below a point-by-point response to all your major comments (our answers in red), in complement to the new manuscript with highlighted main changes (text in red as well).

Sincerely yours,

On behalf of the authors,

Antoine Guillemot

Please also note the supplement to this comment:
https://tc.copernicus.org/preprints/tc-2020-195/tc-2020-195-AC1-supplement.pdf

[Figure]

**Fig. 1.** Figure 16 : (a) Location of the five Miniature Temperature Dataloggers (MTD) that record every hour the temperature of the sub-surface (below 2-10 cm of debris), and (b) the corresponding measurements

[Figure]

**Fig. 2.** Figure 19 : Daily raw (a) and normalized (b) spectrograms of Power Spectral Density (PSD) from OGSA station, located at 2 km from the Laurichard rock glacier, in a stable site, for all the year 2019.

**LAUR 01.DHZ Z**

**LAUR 03.DHZ Z**

**LAUR 04.DHZ Z**

**Fig. 3.** Figure showing normalized spectrograms of other sensors in Laurichard rock glacier, even if data quality is low.

**Supplement:**

**Reply to reviews on the manuscript**
**Modal sensitivity of rock glaciers to elastic changes from spectral seismic noise monitoring and modeling**

By Antoine Guillemot, Laurent Baillet, Stéphane Garambois, Xavier Bodin, Agnès Helmstetter, Raphaël Mayoraz, Eric Larose

Contact :
Antoine GUILLEMOT
ISTerre, Université Grenoble Alpes
Grenoble (France)
*antoine.guillemot@univ-grenoble-alpes.fr*

Dear reviewer, dear editor,

We would like to thank you for the constructive review following the submission of our manuscript "***Modal sensitivity of rock glaciers to elastic changes from spectral seismic noise monitoring and modeling***" to *The Cryosphere*. We took into account all the comments from the two reviewers and editor.

One of the main problems raised concerns the lack of references to the HSVR method. Indeed, we didn't specified clearly that our seismometers are single (vertical) component, so that the HSVR method was inapplicable in our case. However, we added some mentions to previous publications on this method successfully applied on permafrost, since the methodology is very similar to spectral analysis presented in this article.

Furthermore, you suggested additional figures to demonstrate the feasibility of our method: one showing the temporal evolution of modeled resonance frequencies, and another one showing modeled amplitude of the spectrum for a particular state of freezing. Unfortunately, both of them are difficult to address accurately to our mind, because of the lack of information concerning thermo-mechanical coupling and 3D effects on the rock glaciers. We detail these points further in our response.

As suggested by the reviewers, we also modified some sentences and figures in order to improve both the quality of information that we provide and the readability of the publication. We added also some references to publications about seismic monitoring on permafrost and glaciers, as a completed state of art.

Please find below a point-by-point response to all your major comments (our answers in red), in complement to the new manuscript with highlighted main changes (text in red as well).

Sincerely yours,

On behalf of the authors,

Antoine Guillemot

**Comments reviewer 1 (Andreas Kohler)**

 This is a very good and comprehensive study demonstrating the ability of passive seismic measurements for time-lapse monitoring of near-surface structures in combination with other geophysical methods which provide more high-resolution, but time-restricted information. The presented use of modal analysis in particular is a novel and promising approach for rock glacier monitoring. The results presented in this study support the capability if the method.

(1) I would be interested in a brief discussion on how the HVSR (Horizontal-to-Vertical Spectral Ratio) method is related to the modal analysis of single component seismic data. In case of HVSR, the argument often used is that normalizing the horizontal spectrum by the vertical will reduce the source effects and thus enhance the resonance spectrum of the site. HVSR peaks are interpreted either as SH wave resonance (see equation on Page 6, line 190) or maxima of the Rayleigh wave ellipticity in a layered medium. Would it make sense to present the results of this study as HVSR time-lapse spectra instead of single component spectra (assuming the three-component seismometer have been used)? I would expect to see the same temporal variability by potentially reducing at the same time some spectral peaks related to local sources. Is the 3D nature of the rock glacier the reason for not using spectral ratios?

➔ We understand well this comment, because the HVSR method is now commonly used. However, we used only one-component seismometers that measure vertical movement (we then measure only Rayleigh waves, but they are supposed to dominate surface waves). We precise in the new version: "*The seismometers (Mark Products L4C, one vertical component)*". Indeed, we didn't specified clarly that our seismometers were single (vertical) component, so that the HSVR method was inapplicable in our case. However, we added some mentions to previous publications on this method performed on permafrost, since the methodology is very similar to ours.

(2) The ability of the HVSR method for time-lapse permafrost monitoring has been recently investigated in a few studies:

Abbott, R., Knox, H. A., James, S., Lee, R., and Cole, C.: Permafrost Active Layer Seismic Interferometry Experiment (PALSIE), Tech. rep., Sandia National Laboratories (SNL-NM), Albuquerque, NM (United States), available at: https://prod.sandia.gov/techlib/access-control.cgi/2016/160167.pdf (last access: 7 January 2019), 2016.

Kula, D., Olszewska, D., Dobiński, W., and Glazer, M.: Horizontal-to-vertical spectral ratio variability in the presence of permafrost, Geophys. J. Int., 214, 219–231, https://doi.org/10.1093/gji/ggy118, 2018.

Köhler, A. and Weidle, C.: Potentials and pitfalls of permafrost active layer monitoring using the HVSR method: a case study in Svalbard, Earth Surf. Dynam., 7, 1–16, https://doi.org/10.5194/esurf-7-1-2019 , 2019.

 It might be useful to have a look at these studies. Please feel free to not cite them if they are not relevant for the current work (especially since a paper of mine is included).

➔ We were very grateful for sharing these other references, and added some on polar permafrost into our new version. Indeed the HVSR method is broadly used with strong similarities to our spectral analysis, making the parallel relevant. We added this sentence : "*Furthermore, time-lapse monitoring using Horizontal-to-Vertical Spectral Ratio (HSVR) method has already been applied to polar permafrost areas, showing a detectable influence of seasonal variability in the active layer on spectral content of recordings (Köhler and Weidle, 2019; Kula et al., 2018)*."

(3) I miss a figure directly comparing the temporal evolution of the measured resonance frequencies with the modeled once for different states of thawing. Furthermore, a figure comparing the measured and modeled amplitudes spectrum for a particular time would be very useful (for example overlaying figure 2d). In my opinion such figures are more important for demonstrating the reliability of the method than showing the GPR and active seismic results in Fig 4,5 and 6. Those can be moved to the Appendix.

➔ We agree with this comment. This temporal modelling may clearly strengthen the interpretation of our results and improve our publication, but actually we cannot address correctly the transition between seasons. In fact we focus more on the general difference between summer (assuming a complete thawing of the active layer marked by lower values of resonance frequency), and winter (complete freezing of the medium between the surface and the ZAA depth, marked by higher values of resonance frequency). We show the comparison between these extreme values on Figure 13 (winter and summer), but no modelling of the temporal evolution between seasons is proposed. Indeed, a complex thermo-mechanical coupling is required in order to simulate the propagation of the heat wave, but diffusivity properties are poorly known (lack of thermal data from boreholes). In addition, the advection of heat from water adds complexity, and too many assumptions are then required in order to correctly model the time series of resonance frequency using a poroelastic model. Therefore we wouldn't propose this modelling in this publication, even though it should be suggested for future works. The goal of Figure 14 is simply to interpret the relation between ground surface temperature and measured frequencies over the whole year, without any modelling.
The modeled amplitudes spectrum for a particular time would be difficult to compare with the measured one, since 2D and 3D effects on frequency spectrum may appear (see Preiswerk et al. (2019) *(doi: 10.1017/aog.2018.27)*. These effects may be very difficult to assess and to predict for rock glaciers (not studied for now). In this article, we focus on the resonance frequency of the fundamental vibrating mode of the rock glacier (at around 15-20 Hz), rather than on the whole amplitude spectrum. But we specified in the first manuscript the complete methodology of the mechanical modelling performed thanks to Comsol software (shown in Figure 12): the computed Frequency Response Function (FRF) is able to model the fundamental mode of the whole structure, but local 2D and 3D effects may affect this curve, making difficult the comparison between modeled and measured curves.
➔ Concerning the GPR and active seismic figures, they should be moved to the appendix session if the editor will consider that there is too many figures in the main part. Indeed this article focus on passive seismic methods and modal analysis, rather than general geophysics results on rock glaciers.

**Minor comments:**

General: Please check if all commas are needed.

The first sentence of the abstract and the Introduction are identical. Please consider rephrasing.
> ➔ We changed the first sentence of the introduction, as follows: "*Among periglacial landforms, rock glaciers are tongue-shaped permafrost bodies.*"

Line 42: highly challenging?
> ➔ Yes, replaced.

Line 49: Do you mean "coast-intensive"? Or "…remain cost-effective only when limited to one single …"?
> ➔ We specified "*cost-intensive and limited to one single point*", as we mean.

Line 61: active permafrost layer
> ➔ By "surface layers", we want to refer to the active layer, but also the deeper one (until around 10 m depth), which is called permafrost layer since it is composed of permanently frozen materials.

Line 73: Please explain "as a bending beam"
> ➔ We specified : "*as buildings*", in order to suggest a vertical structure fixed at the bottom.

Line 76: "Our goal … of the rock glacier and the time variability of their resonance frequencies which gives hints …"
> ➔ Ok.

Line 78-79: "… are numerically modeled …"
> ➔ Ok.

Line 81: "In the second part, …"
> ➔ Ok.

Line 97: Something is wrong with formatting of exponents
> ➔ Ok.

Line 123: one bracket too much
> ➔ Ok.

Line 130: Are these three-component sensors? Was maintenance required during the measurements (releveling etc.)?
> ➔ We specify in the new version: "*The seismometers (Mark Products L4C, one vertical component)*". For the maintenance required, we already mentioned that the sensors are subject to tilting, forcing to frequent site visits in order to releveling them.

Line 140: I think the time-lapse resonance spectra of the other sensors should be included as supporting materials or in the appendix (do not need to be discussed)
> ➔ We agree. However, since the numbers of figures and appendixes are already high, we share the figures of spectrograms from other sensors (Laurichard sensors C01, C03 and

C04) only in personal communication for the moment (see the figures joined). For us it is not necessary to add them into the article, since these data are not interpretable due to tilting and technical instrumentation issues (specified in the text of the manuscript). Please let us know if you want to include them in an additional appendix.

Line 167: "Continuous seismic monitoring systems are composed …" (?)
➔ We change the verb: "*Continuous seismic monitoring requires autonomous operating systems composed of...*".

Line 171: "… between several sensors and to monitor …"
➔ Ok.

Line 195: I suppose the vertical component is used?
➔ Yes, we used seismometers with only the vertical component. We precise it: "*We pre-processed hourly raw seismic traces from vertical component of seismometers*".

Line 200: This is unclear. Please rephrase to describe how peaks are picked automatically.
➔ Ok, we specified exact values of threshold and tried to explain better the picking : "*We selected automatically significant and sharp peaks of the spectrum by using different threshold values for local maxima picking (minimum of peak frequency at 10Hz, minimum of inter-peak distance at 4 Hz, maximum of width of 8 Hz, minimum of peak height at 0.2 and minimum of prominence at 0.3 for normalized spectra).*"

Line 261: Just to avoid misunderstanding: Is the modelling of resonance frequencies done in full 3D, 2D or for a particular location and 1D model below?
➔ In this article we handle the modelling of rock glacier in 1D for particular location underneath seismic sensors. The 2D and 3D effects are quite difficult to address (see Preiswerk, L.E., Michel, C., Walter, F., Fäh, D., 2019. Effects of geometry on the seismic wavefield of Alpine glaciers. Annals of Glaciology 60, 112–124. https://doi.org/10.1017/aog.2018.27). But we checked that resonance frequency of fundamental mode of the 2D model is similar to the one of our simplified 1D models. We detailed this methodology further (part 4.4).

Line 267: "cost-effective" See above.
➔ We replaced by "*cost-intensive*".

Lines 376, 409, 449, and 452: paragraph/section numbers wrong
➔ Ok.

Line 476: "inter-annual climate variability" Is "climate" the right word here? I guess the constant climate at a particular site includes the inter-annual variability of temperatures.
➔ Yes, we agree with this statement. We then modified the first sentences of this paragraph in order to precise what we handle : "*By tracking resonance frequencies in long term, we are able to detect an inter-annual variability. Interestingly, the freezing process appears to correlate with annual minimum of resonance frequency: as an example, (...)*"

Line 491: "observed gap" Do you mean "observed difference"?
➔ Yes, replaced.

Line 496: "in combination with"?
→ Ok.

Line 498: Remove "Furthermore"
→ Ok.

Line 515: Sentence "Frequency resonance focuses on …": I am not sure if this is correctly formulated. The resonance frequency in general is also an effect of seismic waves propagating through the whole structure. It just depends on the considered frequency band. Here, I agree with your conclusion that resonance works well at high frequencies (and thus for shallower depths) where most of the changes occur, while noise correlation does not work so well due to lack of correlation at high frequencies if the inter-station distance is too large. So, the different sensitivities are mainly because of the nature of the ambient noise wavefield and the sensor network set-up, not because of the depth sensitivity of both methods as such. See for example the study of James et al, where noise correlation could be used to measure very shallow variability with closely located sensors.
→ We specified these statements into the paragraph : "*Furthermore, ambient noise correlation may provide less stable results at high frequencies (up to 14 Hz, for the Gugla study (Guillemot et al., 2020)), preventing any interpretation of the chaotic results due to the lack of high frequency noise in the cross-correlation large inter-sensor distance. Hence, the sensitivity of the different methods depends also on the nature of the ambient noise wavefield together with the sensor network setup. According to the site and its instrumentation, the two passive seismic methods may be combined to obtain stable results along the whole depth of the rock glacier.* "

Line 542: geophysical measurements
→ Ok.

Appendix A: I am not sure if it necessary to include the results of earthquakes since the results are not much discussed. If they are included, one would like to know where the discrepancy compared to noise comes from.
→ We decided to keep this figure in Appendix B, because these results are also used as an argument for another discussion. Indeed, we added a new paragraph (in part. 3.2) to handle the question raised about the noise source variability, and its influence on frequency temporal variability.

Figure 2: Showing the noise waveform record is not necessary in my opinion. Instead, please also add a plot like (d) for Gugla.
→ On our opinion, the Figure 2 eases to show the methodology of frequency picking from a spectrogram of an earthquake. This shows only an example of waveform from an earthquake occurred near the Laurichard rock glacier. The principle for Gugla is strictly the same, therefore a plot for Gugla may overload the figure. In addition, the substantial number of figures and appendixes in this article may be limited, but let us know if you still require this added plot.

---

## Author Comment (AC2) · 23 Oct 2020

Reply to reviews on the manuscript Modal sensitivity of rock glaciers to elastic changes from spectral seismic noise monitoring and modeling

By Antoine Guillemot, Laurent Baillet, Stéphane Garambois, Xavier Bodin, Agnès Helmstetter, Raphaël Mayoraz, Eric Larose

ContactÂă: Antoine GUILLEMOT ISTerre, Université Grenoble Alpes Grenoble (France) antoine.guillemot@univ-grenoble-alpes.fr

Dear reviewer, dear editor,

We would like to thank you for the constructive review following the submission of our manuscript "Modal sensitivity of rock glaciers to elastic changes from spectral seismic noise monitoring and modeling" to The Cryosphere. We took into account all the comments from the two reviewers and editor. One of the main problems raised concerns the influence of ambient noise sources on resonance frequencies that we picked. Indeed, temporal variability of these frequencies could be related with source variability. To address and discuss this relevant question, we kept the figure showing the spectrograms with frequency picking from earthquakes, in order to show that resonance frequency that we picked are still often visible even for variable earthquake sources. Besides, we added a new figure of spectrograms from a station near the Laurichard rock glacier, settled on a stable site that could be considered as a reference station. Furthermore, you suggested additional figures to demonstrate the feasability of our method, showing the temporal evolution of modeled resonance frequencies? Unfortunately, this is out of the scope of this publication to our mind, because of the lack of information concerning thermo-mechanical coupling to accurately address this question. We detailed this point further in our response. Nevertheless, we added a new figure in appendix showing the ground surface temperature data, as you requested. As suggested by the reviewers, we also modified some sentences and figures in order to improve both the quality of information that we provide and the readability of the publication. We added also some references to publications about seismic monitoring on permafrost and glaciers, as a completed state of art. Please find below a point-by-point response to all your comments (our answers in red), in complement to the new manuscript with highlighted main changes (text in red as well).

Sincerely yours,

On behalf of the authors,

Antoine Guillemot

Please also note the supplement to this comment:

https://tc.copernicus.org/preprints/tc-2020-195/tc-2020-195-AC2-supplement.pdf

[Figure]

[Figure]

**Fig. 1.** Figure 16 : (a) Location of the five Miniature Temperature Dataloggers (MTD) that record every hour the temperature of the sub-surface (below 2-10 cm of debris), and (b) the corresponding measurements

**Fig. 2.** Figure 19 : Daily raw (a) and normalized (b) spectrograms of Power Spectral Density (PSD) from OGSA station, located at 2 km from the Laurichard rock glacier, in a stable site, for all the year 2019.

**LAUR 01.DHZ Z**

**LAUR 03.DHZ Z**

**LAUR 04.DHZ Z**

**Fig. 3.** Figure (in annex ?) showing the normalized spectrograms from the other sensors of Laurichard rock glacier, even if data quality is low.

**Supplement:**

**Reply to reviews on the manuscript**
**Modal sensitivity of rock glaciers to elastic changes from spectral seismic noise monitoring and modeling**

By Antoine Guillemot, Laurent Baillet, Stéphane Garambois, Xavier Bodin, Agnès Helmstetter, Raphaël Mayoraz, Eric Larose

Contact :
Antoine GUILLEMOT
ISTerre, Université Grenoble Alpes
Grenoble (France)
*antoine.guillemot@univ-grenoble-alpes.fr*

Dear reviewer, dear editor,

We would like to thank you for the constructive review following the submission of our manuscript "***Modal sensitivity of rock glaciers to elastic changes from spectral seismic noise monitoring and modeling***" to *The Cryosphere*. We took into account all the comments from the two reviewers and editor.

One of the main problems raised concerns the influence of ambient noise sources on resonance frequencies that we picked. Indeed, temporal variability of these frequencies could be related with source variability. To address and discuss this relevant question, we kept the figure showing the spectrograms with frequency picking from earthquakes, in order to show that resonance frequency that we picked are still often visible even for variable earthquake sources. Besides, we added a new figure of spectrograms from a station near the Laurichard rock glacier, settled on a stable site that could be considered as a reference station.

Furthermore, you suggested additional figures to demonstrate the feasability of our method, showing the temporal evolution of modeled resonance frequencies? Unfortunately, this is out of the scope of this publication to our mind, because of the lack of information concerning thermo-mechanical coupling to accurately address this question. We detailed this point further in our response. Nevertheless, we added a new figure in appendix showing the ground surface temperature data, as you requested.

As suggested by the reviewers, we also modified some sentences and figures in order to improve both the quality of information that we provide and the readability of the publication. We added also some references to publications about seismic monitoring on permafrost and glaciers, as a completed state of art.
Please find below a point-by-point response to all your comments (our answers in red), in complement to the new manuscript with highlighted main changes (text in red as well).

Sincerely yours,

On behalf of the authors,

Antoine Guillemot

**Comments reviewer 2**

The study presents a new methodology that uses the resonant frequency of rock glaciers extracted from continuous ambient noise records to monitor the seasonal and interannual changes in the structure of the vibrating glacier. The seismic modal analysis is coupled to a poroelastic model of a 2D porous medium representing the rock glacier, supported by other geophysical measurements. The results indicate the thawing-freezing cycle effects on the resonance of the glacier. This comprehensive study highlights well how the structural changes of rock glaciers can be tracked and monitored with passive seismology, when other geophysical methods are time-consuming and hardly repeatable at such time resolution.

**Major comments:**

(1) I miss somewhere an explanation of the origin of the ambient seismic noise (at frequency > 1Hz). Many studies agree to say that it takes its origin in fluvial processes when the water flow from ice melting in spring/summer creates transient forces on the Earth at the glacier base or on the surrounding ice in englacial channels. The noise recorded in winter may come from other sources in the area. Many studies on ambient noise suggest that a change in the noise sources could lead to a change in the noise spectrum. This renders monitoring studies difficult, especially for ambient noise correlation method. Here you use a modal analysis which should be less sensitive to noise source changes. But still, there remains an open question: are the seasonal variations of the resonant frequencies influenced by changes in the noise or actual structural changes ? I am quite concerned about the abrupt resonant frequency shift you observe at the onset of the melting. Personally, I am absolutely confident in the interpretation of actual changes in elastic properties of the structure underneath the sensor. However, this question should be raised and discussed.

➔ Thanks for this comment, we addressed and discussed this relevant question in the new version. We already studied the noise source variability, because actually melting processes occurring in spring/summer time can affect the nature of ambient noise, and could lead to spurious comparison of spectral content.

➔ During all the year, ambient noise is assumed to be mainly produced by weathering (rain, wind) and anthropogenic sources (traffic on the road in Lautaret Pass for Laurichard, and traffic and human activities in Mattertal valley just below the Gugla rock glacier). In melting and summer periods, noise level is generally higher than during winter time, due to the addition of fluvial and groundwater processes in vicinity of our sites.

➔ We addressed this question by adding a new paragraph, and a figure (Fig. 19) in a new Appendix (C). We compared raw and normalized spectrograms of seismic recordings on rock glaciers and on a stable site located in the vicinity. We then presented the results for the OGSA station, located at Lautaret pass (2 km from Laurichard rock glacier) in 2019 (see Fig. 19). From these raw spectrograms we observed a seasonal variability of noise level, actually suggesting a seasonal variability of the noise sources (higher in summer than in winter, probably due to intensified road traffic and fluvial processes in summer). From the normalized spectrograms, we noticed that no significant changes of frequency peaks appear within the illuminated spectrum of interest (10-40 Hz). Since the OGSA station (stable reference) is located close to the Laurichard rock glacier, we assume that ambient noise sources are the same for both sites. Therefore we conclude

that seasonal variability of frequency peaks recorded on the rock glacier between 15 and 40 Hz (see Figure 3) is not much influenced by source changes, but rather linked to specific resonance of the rock glacier structure.

➔ In addition, we chose to keep Figure 18 in Appendix B, showing spectrograms and frequency picking from several earthquakes signals (hence several sources), in order to show that resonance frequency that we picked are still often visible, even for variable earthquake sources.

➔ We detailed all these observations in the new version: "*The spectral content of seismic recordings can be affected by temporal variations of ambient seismic noise sources. For the two sites, these sources are assumed to originate from stable human activities located in the nearby valley and from weathering, but they could also be partly related to hydrological processes via melting water in spring and summer time. This source variability has to be addressed, in order to eliminate any spurious interpretation of actual changes in elastic properties. We then compare raw and normalized spectrograms of the reference station OGSA over one year (see Fig. 19 in Appendix C) to track any variation of the spectrum content which would prevent further comparison of frequency peaks observed on the rock glacier over time. No significant temporal changes of PSD appears within the illuminated spectrum of this stable station located near Laurichard rock glacier. Another obvious fact to highlight is that frequencies which were picked from ambient noise are also often visible when earthquakes signals are considered (see Appendix B). These two observations strengthen the direct link between these frequency peaks and rock glacier resonance*".

➔ In this way we hope that this question of ambient noise sources variability is now correctly addressed and discussed.

(2) As a proof of concept and to approve the interpretation of the results, I suggest to try to actually reproduce the seasonal variations of the resonant frequencies with the poroelastic model (as stated in the conclusion Line 542 but wrong). This would strengthen the discussion and the capability of the method. The interpretation is finally based on Fig 14 which represents measured resonant frequencies as a function of the temperature of the ground surface, while this temperature does not tell the whole story on what is happening at depth. So having a hint on the evolution of the active layer and the rigidity of the structure thanks to the best-fitting model output would definitely strengthen your conclusions. In addition, this figure would be very great to track the interannual changes. Finally, I miss a figure showing the temperature time-series (in this additional « proof of concept » figure, or in Fig 3 for example). This would ease the reading of Fig 14 and also highlight melting seasons versus winter.

➔ Yes, we agree with this comment. This temporal modelling may clearly strengthen the interpretation of our results and improve our publication, but actually we cannot address correctly the transition between seasons. In fact we focus more on the general difference between summer (assuming a complete thawing of the active layer, marked by lower values of resonance frequency), and winter (complete freezing of the medium between the surface and the ZAA depth, marked by higher values of resonance frequency). We show the comparison between these extreme values on Figure 13. (winter and summer), but no modelling of the temporal evolution between seasons is proposed. Indeed, a complex thermo-mechanical coupling is needed in order to simulate the propagation of the heat wave, but diffusivity properties of rock glaciers are poorly known (lack of thermal data from boreholes). In addition, the advection of heat from water adds complexity, and too many assumptions are then required in order to model the time series of resonance frequency by the poroelastic model. Therefore we wouldn't propose

this modelling in this publication, even though it should be suggested for future works. The goal of Figure 14 is simply to interpret the relation between ground surface temperature and measured frequencies over the whole year, without any modelling. Nevertheless, we decided to add a new appendix with a figure showing the temperature time-series in Laurichard, as you requested.

(3) Sometimes, I think that the structure of the sentences is not smooth enough and lead to confusing statements in English. I have listed some in the comments below which need to be reformulate. In general you should try to keep the sentences short.
➔ Yes, we tried to reformulate and shorten the sentences as possible, together with the corrections of some statements you raised below.

**Other comments:**

L48: in depth -> at depth ?
➔ Ok.

L57-59: References are needed here for both methods (ambient noise vs microseismicity). I understand this is the output of the study by Guillemot et al 2020 as detailed in the following sentence, but other studies have also proven this in glaciated environment.
➔ Yes, we understand the requirement. We completed the references in the new version as follow: "*Passive seismic monitoring systems have the potential to overcome these difficulties on debris slope (Samuel Weber et al., 2018; S. Weber et al., 2018), glaciated (Mordret et al., 2016; Preiswerk and Walter, 2018) and permafrost environments (James et al., 2019; Köhler and Weidle, 2019; Kula et al., 2018), also recently illustrated on the Gugla rock glacier (Guillemot et al., 2020).*"

L89: I suggest to specify « France » after the « Laurichard catchment » Figure 1d: Please indicate the flow direction with an arrow.
➔ Ok.

Section 2.1.3: How were the seismometers maintained (leveling, orientation) ? Are they threecomponent sensors ?
➔ We only used one-component seismometers that measure vertical movement (Rayleigh waves are supposed to dominate surface waves). We precise in the new version: "*The seismometers (Mark Products L4C, one vertical component)*".

You should also specify somewhere here the glacier thickness beneath the stations (ice thickness was also not specified in the previous section)
➔ We guess you addressed the bedrock depth rather than the ice thickness, since rock glaciers are rocky debris bodies. For each sensor and sites (Gugla and Laurichard) we estimated the bedrock depth from different methods (borehole data for Gugla, geophysics for Laurichard) much detailed in the following parts.

L137: « The long periods » -> the longest periods ?
➔ Ok.

L139: What is the distance between these two stations ?
➔ We precise in the text : "*Therefore we decided here to present results from only these two locations, separated by around 80 m*".

L140: It could be interesting to have the results for the seismic measurements at the other stations in the appendix.
➔ Yes, we agree. But since the numbers of figures and appendixes are already high, we share the figures of spectrograms from other sensors (Laurichard sensors C01, C03 and C04) only in personal communication for the moment (see the figures joined). For us it is not necessary to add them into the article, since these data are not interpretable due to tilting and technical instrumentation issues (specified in the text of the manuscript). Please let us know if you want to include them in another appendix.

L144: I suggest to specify « Switzerland » after « Wallis Alps »
➔ Ok.

L157: Please specify here if the geophones are one or 3 components. Are they deployed directly on the ice ? Were they maintained ?
➔ We only used one-component seismometers that measure vertical movement (Rayleigh waves are supposed to dominate surface waves). We precise in the new version: "*The seismometers (Mark Products L4C, one vertical component)*". These sensors are settled on top of relatively large, stable and flat boulders, and sheltered by a plastic tube to shield off any influence of rain, wind and snow. They are connected together with wires insulated by sheath to protect for weather and rockfalls. All these details about instrumentation are in the manuscript.

L169-171: This is the right place to be more specific for glacier microseismicity and glaciological applications (location of crevasses, basal interface and asperities, water channels) with appropriate referencing. For ambient noise studies on glaciers, you could refer to the studies of Preiswerk and Walter 2018, Sergeant et al 2020 (for the imaging part) and Lindner et al 2018 (for the monitoring part)
Preiswerk, L. E., & Walter, F. (2018). High-Frequency (> 2 Hz) Ambient Seismic Noise on High- Melt Glaciers: Green's Function Estimation and Source Characterization. *Journal of Geophysical Research: Earth Surface*, *123*(8), 1667-1681.

Sergeant, Amandine, et al. "On the Green's function emergence from interferometry of seismic wave fields generated in high-melt glaciers: implications for passive imaging and monitoring." *The Cryosphere* 14.3 (2020): 1139-1171.

Lindner, F., Weemstra, C., Walter, F., & Hadziioannou, C. (2018). Towards monitoring the englacial fracture state using virtual-reflector seismology. *Geophysical Journal International*, *214*(2), 825-844.
➔ Thanks a lot for sharing these references. We added several of them in order to complete the former state of art: "*Passive seismic monitoring systems have the potential to overcome these difficulties on debris slope (Samuel Weber et al., 2018; S. Weber et al., 2018), glaciated (Mordret et al., 2016; Preiswerk and Walter, 2018) and permafrost environments (James et al., 2019; Köhler and Weidle, 2019; Kula et al., 2018)*" But since rock glaciers behave rather like rocky landslides than glaciers, we wouldn't favor references on glaciological applications rather than others. We focus more on references

about spectral analysis of ambient noise recordings on glacial and periglacial environments.

L171: add « to » before monitor.
➔ Ok.

L174: « on it » -> « on site »
➔ Ok.

L176: « tracking dynamic parameters » -> « tracking the evolution of »
➔ Ok.

L180: « The seismic noise averaging properties » What does this refer to ?
➔ We modify the sentence to precise what property is in question: "*Through stacking source and trajectories over time and space, seismic noise allows considering the illuminated frequency spectrum as large and stable enough to overcome these respective effects, particularly when monitoring is considered.* "

L182: « The PSD is simply defined by averaging the intensity of the FFT » You average it by what ? The (inverse of) time of integration ? I do not see any averaging in the proposed equation.
➔ We wanted to mention that we stacked the seismic frequency content over time (hourly or daily) before computing the PSD ; that's why we considered a time-averaging. But we already precised that we process hourly raw data in the previous version, so that we remove this mention here : "*The power spectrum density (PSD) is simply defined by computing the intensity of the fast Fourier transform (FFT) of the seismic record*".

L185. Preiswerk et al 2019 also investigated the resonant glaciers with different geometries which imply 1D, 2D and 3D resonances through HVSR, time-frequency dependent polarization and modal analysis. This study should be cited, here maybe or elsewhere.

Preiswerk, L. E., Michel, C., Walter, F., & Fäh, D. (2019). Effects of geometry on the seismic wavefield of Alpine glaciers. *Annals of Glaciology*, *60*(79), 112-124.
➔ Thanks for sharing these relevant reference. We actually mentioned it twice in the new version (in methods of spectral analysis of seismic data (part. 3.1) "*The estimation of bed geometry properties from spectral analysis of seismic noise has already been studied on alpine glaciers  (Preiswerk et al., 2019)*", and modal analysis and frequency response of the rock glacier (part 4.4) "*Similar to the fundamental mode of an unstable rock mass (Burjánek et al., 2012) and avalanche glaciers (Preiswerk et al., 2019), the measured polarization is almost linear*".

L191: « in absence of geometrical changes » maybe specify here « (i.e. ablation/accumulation) »
➔ Yes, we agree. In a case of rock glacier, the potential geometrical changes may be related rather to an intensive creeping movement (or a destabilization), than to accumulation or ablation (like glaciers). Then we precise this with accurate words for rock glaciers: "*Extending this approach to a rock glacier shows that in absence of*

*geometrical changes (no significant destabilization), resonance frequency variations can be related to evolution of its rigidity, through Young's modulus and density*".

L191: missing comma after Young's modulus.
➔ Ok.

L195: Which component ?
➔ We precise in the new version: "we pre-processed hourly raw seismic traces from vertical-component seismometers".

L200-204: The picking method is not clearly described. Additionally, please specify why you expect to have frequency peaks above 10 Hz on the glacier ? Is it related to the glacier morphology and more specifically to the ice thickness ?
➔ We rewrite this part to clarify some points. We precise the values of threshold : "*We selected automatically significant and sharp peaks of the spectrum by using different threshold values for local maxima picking (minimum of peak frequency at 10Hz, minimum of inter-peak distance at 4 Hz, maximum of width of 8 Hz, minimum of peak height at 0.2 and minimum of prominence at 0.3 for normalized spectra)*."
➔ We don't expect to see any resonance frequency of the rock glacier below 10 Hz, after viewing the spectrograms of all sensors on the two sites. Furthermore, for a simplified 1D model with soft rock glacier on hard bedrock, the frequency of the fundamental mode would be f0=Vs/4h (mentioned in the article). With h= 10m and Vs=600m/s (orders of magnitude), peaks below 10 Hz would be unlikely for specific resonance of the whole rock glacier body.

Figure 2d: It seems that you still see the 23 Hz anthropogenic peak at station C05 (as expected) but it is not obvious because of the normalization. Maybe indicate this in the caption.
➔ Yes, we indicated this in the caption.

L212: « The results of PSD » -> « The spectrograms of PSDs »
➔ Ok.

L218: « no subglacial resonating water-filled cavities was known on the site ». I suspect you refer to moulins which extend from the glacier surface to (probably) the glacier base (this is what it sounds as you cite Röösli et al 2016) ? If yes, the part of the sentence Line 2018 should me modified to point out to moulins. If no, how do you know that there is absolutely no channels in the glacier ?
➔ Yes, we agree with that. We then modified the sentences to clarify the two hypotheses (hydrological because groundwater can exist, even if depth and location are unknown, and anthropological) : "*Since this frequency peak is fully stable over time, we interpret its origin as either hydrological or anthropogenic. It may be generated either by groundwater flow within the rock glacier (Roeoesli et al., 2016), or by a pressure pipe located 400 m downstream or from road traffic coupled with a tunnel near the Lautaret pass (see Fig. 1). This frequency peak is also visible on spectrograms of station OGSA (see black arrow in Fig. 2d) located at Col du Lautaret on a stable site, suggesting a potential anthropogenic source. The spectral content of these recordings exhibits the same peak at 23-24 Hz (see red curve in Fig. 2d), implying it is not directly related to*

*the rock glacier resonance. Then this frequency peak is hereafter excluded from the analysis".*

L225: « resonating structure of the sensor » -> « beneath the sensor » ? Do you refer to the glacier or the shelter around the sensor ?

➔ We referred to the shelter around and above the sensor (precise in the new version).

Section 3.2: see major comment: I miss somewhere an explanation of the origin of the ambient seismic noise (at frequency > 1Hz). Many studies agree to say that it takes its origin in fluvial processes when the water flow from ice melting in spring/summer creates transient forces on the Earth at the glacier base or on the surrounding ice in englacial channels. You say later (Line 224) that water filling of the resonating structure could be responsible for changes in PSD. Then you say Line 229 that no water was present in the sensor settlement. In fact, the presence of meltwater inside the glacier does affect the seismic noise with strong noise generated in spring/summer and little englacial noise in winter. The noise recorded in winter may come from other sources in the area. This should appear on non-normalized spectrograms. In this case, there remains an open question on the cause of the resonating frequency shift from winter to summer. Is it related to source effects or structural changes ? I am confident in the interpretation of actual changes in elastic properties of the structure underneath the sensor. However, this question should be raised at the end of this section or in the discussion.

➔ Yes, we address this question (see major comment above). The new paragraph that we added is : "*The spectral content of seismic recordings can be affected by temporal variations of ambient seismic noise sources. For the two sites, these sources are assumed to originate from stable human activities located in the nearby valley and from weathering, but they could also be partly related to hydrological processes via melting water in spring and summer time. This source variability has to be addressed, in order to eliminate any spurious interpretation of actual changes in elastic properties. We then compare raw and normalized spectrograms of the reference station OGSA over one year (see Fig. 19 in Appendix C) to track any variation of the spectrum content which would prevent further comparison of frequency peaks observed on the rock glacier over time. No significant temporal changes of PSD appears within the illuminated spectrum of this stable station located near Laurichard rock glacier. Another obvious fact to highlight is that frequencies which were picked from ambient noise are also often visible when earthquakes signals are considered (see Appendix B). These two observations strengthen the direct link between these frequency peaks and rock glacier resonance.*"

➔ ". We also added a new Appendix (B) with a figure to show and clarify this point.

Figures 3/4: The blue boxes indicating the melting season are present only for spring. On what observations is it based on ? Generally in Alpine glaciers, the melting seasons spread from spring (April/May) to the end of summer (september/october).This would also be consistent with the seasonal cycle you observe for the seismic dominant frequency. Also related to this comment, I would say in Line 233: « a sudden drop of frequency occurs at the time when melting processes [start to occur in spring and stay stable to lower frequency over the course of the summer]. »

➔ Melting periods are defined based on thermal data acquired on thermistors located at the ground (around 10 cm depth) surface around the rock glacier. They are limited to the period when the "zero-curtain effect" (temperature constant and equals to the

freezing point) is visible, which highlights the partial refreezing of melting water. We added a new figure showing these data in Appendix (Figure 16) in order to clarify this point. When the temperature goes up to positive values, we consider that the melting period is finished, since active layer is totally thawed until the surface. But of course, melting water from upstream still remains, extending the "melting period" to the end of summer, in a sense that you suggest.

Figures 3-4: I would here display the time series of air or ground temperature + periods of snow cover. This would ease the interpretation in the discussion and Fig 14
➔ Yes, we decided to add a new appendix with a figure showing the temperature time-series in Laurichard (Figure 16), as you requested.

Figure 4: Nice observation !

Line 247: I would add « at [spring and] summer time »
➔ Ok.

Line 261: Do you have a reference for the software ?
➔ Yes, we added a reference as bottom of page : " COMSOL Multiphysics® v. 5.4, www.comsol.com, ComsolLab, Stockholm, Sweden."

Figures 5-7: I suggest to move these figures to the appendix section as I think that the study should be focused on the modal analys/modeling methodology.
➔ Ok, we agree. These figures should be moved to the appendix session if the editor will consider that there is too many figures in the main part. Indeed this article focus on passive seismic methods and modal analysis, rather than general geophysics results on rock glaciers.

Line 391: « appling » -> applying
➔ Ok.

Line 446: « as shown in Fig 13 for Laurichard and not presented here but similar for Gugla » Results for both glaciers are presented in Fig 13, so you should revise this sentence.
➔ Ok, we removed this sentence.

Line 449: Section number missing (and elsewhere)
➔ Ok, checked.

Line 450: « and compare them to the maximum values of the observed ones ». You should indicate here that data are indicated by squares and rephrase this as the text alone is confusing. Say that you represent here the maximum and minimum bounds of the resonance frequencies you measure for each mode, for the two years - if I understand correctly.
➔ Yes, you understand correctly the Figure 13. But we clarify the text to prevent confusion : "*We compare them to the maximum (in winter) and minimum (in summer) values observed on all sites (depicted in squares in Fig. 13, values from Fig. 3 for Laurichard and Fig. 4 for Gugla).* "

Line 452-453: « Resonance frequencies of these modes match the freq band of measurements below 50 Hz, and generally decrease with thawing ». There is no relation between the two parts of this sentence so you should split it in two. The first part is a bit redundant with the previous sentence of the text. In general I think that section 4.5 could be better rewritten and reorganized with clearer description of the glacier seismic behavior.

➔ Yes, we reorganized a bit this paragraph. This sentence has been moved before, as it deals with general statements about expected resonance frequency.

Line 454: I would add at the end after « for all cases » « i.e. Gugla and Laurichard »

➔ Ok, added.

Line 455: You should remind here that C05 is towards the glacier tongue (on thinner ice H=8m according to your velocity model) and C00 is more upstream (H=14 m).

➔ It is not about ice thickness (unknown here), but rather about bedrock depth. This major difference between sensors C00 and C05 is reminded in this paragraph.

Figure 13: You could add arrows: one which points toward higher depths of thawing indicating summer, one which points toward 0 indicating winter

➔ Ok, we modified Figure 13 as requested.

Line 465: I would replace « melting periods » by « at the onset of the melting period in spring » (summer is also a melting period).

➔ Yes, we replaced this expression by: "*a sudden drop at the onset of melting periods in spring and lower values during summers*".

Line 471: « observed seasonal freq variations » -> observed freq seasonal variations

➔ Ok.

Lines 487-488: « in 2019 … frequency is lower than is 2018 » Give values !!

➔ We precise: "*Similarly for the Gugla site, the winter resonance frequency was significantly lower in 2017 (19 Hz)  than in the others years (around 23 Hz).*"

Line 488: « In addition to an earlier snow cover period in 2019 than 2018 … » I would reformulate. Did snow falls started earlier in 2019 or was the period for snow cover shifted earlier in time ? « frozen » -> « refrozen » ?

➔ We reformulated : "*earlier and longer snow cover in 2019 than in 2018*".

Lines 475-489: I think this paragraph could be improved with some reorganization. You should highlight your conclusion which is that, on average, resonance frequencies are more sensitive to the intensity of internal thawing (which is influenced by …) than to the air temperature as … Also you use both the present and the past to describe the observations. You should homogenize this.

➔ We homogenized this paragraph and tried to clarify the examples together with the main message in conclusion.

Could you explain those differences with your model ? Can different level of porosity and depth of thawing mimic different intensity of thawing ? It would be very great to have a figure

showing the time series of the observed resonance frequencies with the model results for different states of thawing.

➔ This request is already discussed as a major comment above. This temporal modelling may clearly strengthen the interpretation of our results and improve our publication, but actually we cannot address correctly the transition between seasons (which state of freezing/thawing correspond to which time ?). We focus more of the extreme values (low values in summer, high values in winter), as depicted in the modified Figure 13.

Lines 489-498: I suggest to remove this paragraph from the discussion which is focused on the thawing-freezing cycle.

➔ We agree that the discussion deals mostly with freezing-thawing cycle, but we wanted to highlight the fruitful benefit of GPR results on the bedrock depth estimation, in combination with passive seismic measurements. Indeed, the difference of bedrock between the two sensors for Laurichard is the unique reason for the gap of resonance frequency amplitude between them. Thus, an accurately estimated bedrock depth is required to finely analyze and model the effect of seasonal thermal forcing on our seismic measurements. Let's us know if you still want to remove this paragraph from the discussion, in this case it can be moved to the appendix section.

Line 499: I would remove « non-linear ».

➔ Ok.

Line 500: « over the year » -> along the year

➔ Ok.

Line 501: I would remove « dry » as we do not know if you are referring to the air or the rock glacier.

➔ Ok.

Lines 521-522: « for ambient noise correlation method, the theoretical relative velocity change of the Rayleigh wave is computed by dispersion curve difference using the Geopsy package» -> I would say « for … corr method, we compute the dispersion curves of the Rayleigh wave using the Geopsy software. The theoretical relative velocity changes are computed by measuring the differences of the dispersion curve with the reference one at each frequency. »

➔ We reformulated: "*for ambient noise correlation method: we compute the dispersion curves of the Rayleigh waves using the Geopsy package (Wathelet et al., 2004). The theoretical relative velocity changes (dV/V) are computed by measuring the difference between the modified dispersion curve with the reference one at each frequency*".

Figure 15: For the modal analysis, you say « first mode » while everywhere in the text you refer to the fundamental mode. This could be confusing. You should specify the resonance frequency, and also in the main text.

➔ Yes, we changed the "first mode" to "mode 0". Mode 0 is already mentioned in Figure 13 and is commonly used for referring the fundamental mode.

Line 538: I would say « continuous seismic noise measurements »

➔ Ok.

Line 539: This is minor but I would say something like « These freq show seasonal variations that are to be related with changes in elastic properties of the structure underneath the recording sensor, due to freeze-thawing effects ».

➜ Ok, we modified by: "*These frequencies show seasonal variations, related with changes in elastic properties of the structure underneath the recording sensor, due to freeze-thawing effects*".

Line 541: « which fit well the recorded frequencies » I would specify here or next sentence on what the resonance freq depend in the poroelastic model. Basically the take-home message I keep from your analysis is that the freq depend on the maximum depth of thawing in a porous medium.

➜ For us this sentence is a general statement for the good agreement between modeled and measured extreme values of resonance frequency. We detailed below this take-home message (see next comment).

Line 542: « we have reproduced the observed seasonal variations » This is a strong statement, you did not reproduced the seasonal cycle exactly but only reproduce the maximum freq in winter due to maximum freezing and minimum freq in summer due to maximum thawing. See my major comment.

➜ Yes, we precise : "*we have reproduced the observed higher values in winter due to maximum freezing, and lowest values in summer due to maximum thawing*".

Line 551: « insight to other deeper processes » -> into other processes at greater depth

➜ Ok, replaced.

---

## Author Response (AR2)

**Reply to reviews on the manuscript**
**Modal sensitivity of rock glaciers to elastic changes from spectral seismic noise monitoring and modeling**

By Antoine Guillemot, Laurent Baillet, Stéphane Garambois, Xavier Bodin, Agnès Helmstetter, Raphaël Mayoraz, Eric Larose

Contact :
Antoine GUILLEMOT
ISTerre, Université Grenoble Alpes
Grenoble (France)
*antoine.guillemot@univ-grenoble-alpes.fr*

Dear reviewer,

We wish to thank you for the constructive review following the submission of our manuscript "*Modal sensitivity of rock glaciers to elastic changes from spectral seismic noise monitoring and modeling*" to *The Cryosphere*. We took into account all your comments, and wish that our manuscript is improved accordingly.

Since the editor decided that only minor revisions would be required before the manuscript can be accepted, we only changed the technical details that you raised. For example, we addressed the previous request to reduce the large number of figures in the main text by partly moving them in appendices. We also adapted the numbering notation of figures according to *TC* requests.

Please find below a point-by-point response to all your comments and those from the editor (our answers in red), in complement to the new manuscript with highlighted main changes (text in red as well).

Sincerely yours,

On behalf of the authors,

Antoine Guillemot

**Report 1 (anonymous referee 2)**

**Suggestions for revision or reasons for rejection (will be published if the paper is accepted for final publication)**

I appreciate the changes brought by the authors to the presented article. The authors brought sufficient argument to the two issues I pointed out: i.e. the noise source variability and the temporal evolution of the resonant frequency. They also point out the differences expected in ambiant noise analysis from a glacier and a rock glacier, which was a bit confusing for me at the first reading of the paper. I think the discussion is well organized now and I am satisfied with the study.

The authors claim that a full modelling of the time evolution of the resonant frequency may be hard to address for the transition season, which I completely understand. This is a bit frustrating, but the authors argument in this way in the new version of the manuscript.

The naming of the figures were a bit confusing to me wether they are part of the main article or they are moved to the appendix or supplementary material. I would suggest to keep in the main article figures 1 to 4, 8 to 15; move to the appendix Figures 5, 16 to 19; move to the supplementary material Figures 6 to 7. Also for figure 5, it is unusual to split the figure labelling in Fig5:1(a-b) and Fig:2(a-b). Maybe use Fig 5a-d. It will be easier to reference the figure in the main text.

> ➢ The number of figures was actually confusing. We decided to move Figures 5, 6 and 7 to Appendix D (with new notation D1, D2, D3 respectively). We chose to keep Figures 6 to 7 to Appendix D because they show results from geophysical surveys in Laurichard rock glacier, and thus they are part of this appendix for us. But if the number of figures in appendices is also too high, we totally agree to move Figures 6 and 7 to Supplementary Materials in the final version.
> ➢ We also renamed Figures 16 to 19 to Figures A1, B1, B2, C1 respectively, according to the *TC* notation.
> ➢ Finally we modified the labelling of Figure 5 (a-d) accordingly.

**Other comments (from the editor)**

[Lines 16 - 17]
I suggest to re-write this phrase (by correcting punctuation and making it less wordy) as:

"Here, we analyse the spectral content of ambient noise to study the modal sensitivity of rock glaciers, which is directly linked to the system's elastic properties."
- ➢ Ok, we modified.

[Line 20]
hot periods -> "warm periods" seems to be more appropriate for the temperature range described.
- ➢ Ok

[Line 40]
high resolution observations
-> {add a hyphen}
high-resolution observations
- ➢ Ok

[Line 50]
here and elsewhere, please use
[e. g.] as [e.g.,] {with a comma}
- ➢ Ok

[Lines 54-56]
This phrase is difficult to follow due to its length, and I suggest to split it into two for clarity.

"Passive seismic monitoring systems have the potential to overcome these difficulties on debris slope (Samuel Weber et al., 2018; S. Weber et al., 2018), glaciated (Mordret et al., 2016; Preiswerk and Walter, 2018) and permafrost environments (James 55 et al., 2019; Köhler and Weidle, 2019; Kula et al., 2018), also recently illustrated on the Gugla rock glacier (Guillemot et al., 2020)."
- ➢ We agree. We split it : "Passive seismic monitoring systems have the potential to overcome these difficulties on debris slope (Samuel Weber et al., 2018; S. Weber et al., 2018), glaciated (Mordret et al., 2016; Preiswerk and Walter, 2018) and permafrost environments (James et al., 2019; Köhler and Weidle, 2019; Kula et al., 2018). The interest of such method has recently been illustrated also on the Gugla rock glacier (Guillemot et al., 2020)."

[Line 87]
Presentation of the sites
-> {I suggest more straightforward}
Study sites
- ➢ Ok

[Line 104]
The long term survey
->
The long-term survey
> Ok

[Line 105]
one meter, (Marsy et al., 2018)).
-> no need for the comma.
> Ok

[Line 107] here and elsewhere
at around 1m/yr
-> {around is colloguial}
at approximately 1_m/yr
> Ok

[Line 118]
ERT surveys
- ERT was defined so far.
> Ok, we precised : "Electrical Resistivity Tomography (ERT)"

[Line 134]
please, indicate the eigenfrequency of these sensors
> Ok, we precised : "Mark Products L4C, one vertical component, eigenfrequency 1 Hz"

[Lines 138-141]
usage of "although" in this phrase seems awkward. Please rephrase for your intended meaning. "...and despite our frequent site visits for sensor releveling"?
> Ok, we rephrased : "and despite our frequent site visits for sensors releveling"

[Line 139] if you stick to
in order to releveling sensors
->
... to relevel sensors
> Ok

[Line 162]

geophysical campaign from seismic refraction
->... {for} seismic refraction?
  ➢ Yes, "for".

[Line 174]
variations (Snieder ... 2015, for a review).
->
... (for a review, see Snieder ... 2015).
  ➢ Ok

[Line 290]
the paragraph has an unnecessary indent.
  ➢ Ok, corrected.

[Line 306]
I do not understand what is meant by "(see 0)". Please check.
  ➢ Sorry for this bug. We removed this unnecessary mention.

(Figure 5(2))
As the referee also pointed, TC does not use such referencing to Figures, and it should be corrected here, in the caption, and the figure labels.
  ➢ Ok, corrected.

[Line 328]
The Vs profiles displayed in Figure 7b shows
-> show {plural}
  ➢ Ok.

best fitting models
-> best-fitting models
  ➢ Ok.

[Line 362]
bedrock, (Hausmann et al., 2012)),
-> no need for the comma.
  ➢ Ok.

[Line 368]
(see 0 for Laurichard and 0 for Gugla)
"see 0"? I do not follow this.
  ➢ Sorry for this bug, we modified : "see Figure 8(c) for Laurichard and Figure 9 for Gugla "

[Line 375]
((CREALP, 2016)),
remove redundant parenthesis
> Ok.

[Line 440]
(i.e.
->
(i.e.,
> Ok.

[Line 493]
temperature reaches 0C and stay
- stays {plural}
Please, also recheck your intended meaning. I am not sure what do you mean by temperature stays during a zero-curtain period; something is missing.
> Yes, we precised : "when temperature reaches 0°C and stays constant during a zero-curtain period"

[Line 530]
a 2D mechanical modeling of rock glaciers, which fit
->
... which fits {plural}
> Ok.

[Line 535]
investigations, as
->
investigations, {such} as
> Ok.

Line 549
CREALP (see weblink in references)
-I failed to find such a link. Please check.
> Sorry the reference link was missing. We then added the link in the references : "www.vs.ch/programme-pilote-ofev-cryosphere"

[Line 606] and similar below
Fig5(1b)- see my earlier comment
> Yes, we adapted the labelling of the figure.

Line 885
(see 1b)
- I could not follow where and what this is.
> We removed this unnecessary mention.

Fig 12
label (c) drifted to the left subplot (b) and has to be placed appropriately.
> Yes, modified in the figure.